# Differential impact of ubiquitous and muscle dynamin 2 isoforms in muscle physiology and centronuclear myopathy

Raquel Gómez-Oca [1,2], Evelina Edelweiss[1], Sarah Djeddi[1], Mathias Gerbier[2], Xènia Massana-Muñoz[1], Mustapha Oulad-Abdelghani[3], Corinne Crucifix[4], Coralie Spiegelhalter[3], Nadia Messaddeq[3], Pierre Poussin-Courmontagne [4], Pascale Koebel[3], Belinda S. Cowling [2,5] ✉ & Jocelyn Laporte [1,5] ✉

Dynamin 2 mechanoenzyme is a key regulator of membrane remodeling and gain-of-function mutations in its gene cause centronuclear myopathies. Here, we investigate the functions of dynamin 2 isoforms and their associated phenotypes and, specifically, the ubiquitous and muscle-specific dynamin 2 isoforms expressed in skeletal muscle. In cell-based assays, we show that a centronuclear myopathy-related mutation in the ubiquitous but not the muscle-specific dynamin 2 isoform causes increased membrane fission. In vivo, overexpressing the ubiquitous dynamin 2 isoform correlates with severe forms of centronuclear myopathy, while overexpressing the muscle-specific isoform leads to hallmarks seen in milder cases of the disease. Previous mouse studies suggested that reduction of the total dynamin 2 pool could be therapeutic for centronuclear myopathies. Here, dynamin 2 splice switching from muscle-specific to ubiquitous dynamin 2 aggravated the phenotype of a severe X-linked form of centronuclear myopathy caused by loss-of-function of the MTM1 phosphatase, supporting the importance of targeting the ubiquitous isoform for efficient therapy in muscle. Our results highlight that the ubiquitous and not the muscle-specific dynamin 2 isoform is the main modifier contributing to centronuclear myopathy pathology.

Dynamins are large GTPases and main regulators of membrane fission and cytoskeleton re-organization. In recent years, dynamins were implicated in several diseases with tissue-specific phenotypes and were proposed as therapeutic targets[1–3]. The physiological role of dynamins in specific tissues are not well characterized. Here we focus on the ubiquitously expressed dynamin 2 (DNM2) and characterize the skeletal muscle-specific function of its different isoforms and their modulation as a therapeutic target in a congenital myopathy.

Dynamins define a large family of GTPases including classical dynamins and dynamin-related proteins, conserved through evolution in Eukaryotes including yeast and plants [4]. Dynamin binds lipids with its pleckstrin homology (PH) domain, oligomerizes through the middle/stalk domain as a higher-order helical structure, and hydrolyzes GTP concomitantly with membrane fission[5–7]. The PH-stalk interface in the closed conformation is auto-inhibitory and release of the PH domain upon lipid binding leads to an open conformation[8]. Dynamins

[1]Dpt Translational Medicine, Institut de Génétique et de Biologie Moléculaire et Cellulaire (IGBMC), INSERM U1258, Université de Strasbourg, CNRS UMR7104 Illkirch, France. [2]Dynacure, Illkirch, France. [3]Core platforms, Institut de Génétique et de Biologie Moléculaire et Cellulaire (IGBMC), INSERM U1258, Université de Strasbourg, CNRS UMR7104 Illkirch, France. [4]Integrated Structural Biology platform, Institut de Génétique et de Biologie Moléculaire et Cellulaire (IGBMC), INSERM U1258, Université de Strasbourg, CNRS UMR7104 Illkirch, France. [5]These authors contributed equally: Belinda S. Cowling, Jocelyn Laporte. ✉e-mail: Belinda.cowling@dynacure.com; jocelyn@igbmc.fr

were first identified as microtubule-associated proteins, and bundle actin filaments[9,10]. Classical dynamins are mechanoenzymes that fission membranes at different localizations in the cell. They are thus main regulators of clathrin-mediated endocytosis[11], formation of vesicles from the trans-Golgi network[12], autolysosome reformation and lipid droplet fission[13]. There are 3 classical dynamins, DNM1 is mainly expressed in brain, DNM3 in brain, lung and testis, and DNM2 is ubiquitously expressed. Different splice isoforms can have different cellular functions[14]. In *DNM2*, exons 10a and 10b are mutually exclusive exons, while the in-frame exon 13b encoded between exons 13 and 14 is alternatively spliced. Differential splicing was found to impact Golgi targeting of DNM2[15,16]. More recently an in-frame alternative exon 12b was found in skeletal muscle and encodes for 10 amino acids located between the PH and the middle/stalk domains close to the auto-inhibitory interface[17]. The functional and physiological importance of this isoform is not yet known.

*DNM2* is mutated in different neuromuscular diseases: autosomal dominant centronuclear myopathy (CNM; MIM#160150)[18,19], autosomal dominant Charcot-Marie-Tooth peripheral neuropathy (CMT; MIM #606482)[20], and recessive lethal congenital contracture syndrome (MIM#615368)[21]. The majority of CNM mutations are located in the auto-inhibitory interface, while the CMT mutations mostly concentrate in the lipid binding loops of the PH domain[5,6]. Despite its ubiquitous expression, DNM2 is thus implicated in tissue-specific diseases. *DNM2* mutations in CMT correlate with axonal or myelin neuronal defects leading to progressive muscle weakness. Mutations in CNM cause abnormal organelle positioning in myofibers including nuclei centralization, sarcoplasmic reticulum and mitochondria mis-position, and myofiber hypotrophy, leading to proximal and facial muscle weakness. Severe forms of CNM appear at birth with congenital hypotonia and the full spectrum of histopathological and clinical signs[22]. Conversely, milder adult-onset forms may not display the full histological hallmarks and can be associated to necklace myofibers, i.e. sub-sarcolemma basophilic rings composed of mitochondria, reticulum and glycogen[23,24].

The role of DNM2 in skeletal muscle is not well defined. Studies in drosophila, zebrafish and mouse suggested a potential role in T-tubules formation or maintenance[25–27], autophagy[28,29], and lipid droplets and mitochondria metabolism[30]. Defects or mutations in *DNM2* also correlated with alteration of the neuromuscular junction[31–33].

Other forms of CNM with variable severity and similar histopathological and clinical hallmarks exist[34]. X-linked CNM (XLCNM, also called myotubular myopathy) is caused by mutations in the phosphoinositide phosphatase myotubularin (*MTM1*; MIM#310400) and autosomal dominant and recessive CNM due to mutations in the DNM2 partner amphiphysin 2 (*BIN1*; MIM#255200). Based on the hypothesis that these genes are in the same cellular pathway of membrane remodeling and that DNM2 gain-of-function is a common pathomechanism for several CNM forms, downregulation of DNM2 was proposed as a therapeutic strategy. Downregulation of all DNM2 isoforms (pan-DNM2) efficiently prevented and reverted CNM linked to *MTM1*, *BIN1* and *DNM2* mutations in mouse models[16,17,31,35–38]. In these studies, the total pool of DNM2 was reduced.

Here, we studied the role of the different DNM2 isoforms in healthy skeletal muscle, in centronuclear myopathy, and the therapeutic mechanism of DNM2 targeting in CNM. We then tested if targeting specifically muscle DNM2 through removal of exon 12b is efficient to rescue the CNM hallmarks of the *Mtm1*$^{-/y}$ mouse, a faithful model of XLCNM, to increase the specificity of this therapy towards the muscle tissue.

## Results

### Inclusion of exon 12b defines the main muscle-specific DNM2 isoforms

Known DNM2 alternative in-frame exons are 10a and 10b (mutually exclusive exons) and exon 12b and 13b (Fig. 1a). Here we named the ubiquitous isoforms lacking exon 12b Ub-*Dnm2*, and the isoforms including exon 12b M-*Dnm2*. The total pool of *Dnm2* RNA (Ub + M-*Dnm2*) is referred to as pan-*Dnm2*. Exon 12b encodes 10 amino acids and 7 out of 10 amino acids are conserved between mouse and human (Supplementary Fig. 1a). Primers specific for M-*Dnm2*, Ub-*Dnm2* and pan-*Dnm2* were validated by RT-qPCR and a rabbit antibody was raised against the exon 12b peptide and detects specifically mouse and human M-*Dnm2* (Fig. 1a, Supplementary Fig. 1b).

RNAseq analysis of tibialis anterior (TA) muscles from wild-type mice before and after birth revealed that M-*Dnm2* encompassed 21% of all pan-*Dnm2* at E18.5 and increased to 44% and 49% at 2 weeks (wks) and 7wks of age, respectively (Fig. 1b). M-*Dnm2* expression levels mainly increased during the first 2 wks, corresponding with muscle hypertrophy and maturation of the intracellular organization of myofibers[39,40] (Fig. 1c). Transcriptomic analysis revealed that M-*Dnm2* isoforms include the mutually exclusive exon 10a or 10b but never 13b (Supplementary Fig. 1c). The 2 most expressed *Dnm2* transcripts in adult mouse muscle are the Ub-*Dnm2* including exon 10a and the M-*Dnm2* including exon 10a. M-*Dnm2* also increased during myoblast differentiation at the RNA and protein levels (Supplementary Fig. 1d, e). M-*Dnm2* is mainly expressed in skeletal muscle and detectable in heart at the RNA and protein levels, while we confirmed Ub-*Dnm2* is expressed in all tissues tested (Fig. 1d, e; Supplementary Fig. 1f). Similar data were reported in the GTEx expression database for human (www.gtexportal.org) and we specifically confirmed this in human skeletal muscle, where M-*DNM2* represented 44% of the total pool of *DNM2* (Supplementary Fig. 1c). Taken together, these results show that M-*Dnm2* is mainly expressed in skeletal muscle and increased during postnatal muscle maturation.

### M-*Dnm2* is dispensable for muscle development and function

To gain insight into the muscle-specific function of DNM2, we constitutively deleted the in-frame exon 12b in mouse (Supplementary Fig. 2a). *Dnm2*ex12b$^{-/-}$ mice are viable up to at least 1.5 years (yrs) of age (Fig. 2a), and the proportion of genotypes obtained was equivalent to the expected Mendelian ratio (Supplementary Fig. 2b; Supplementary Table 1). RNA and protein analyses from muscle validated the absence of M-*Dnm2* in different muscles while pan-*Dnm2* was slightly but significantly increased (Fig. 2b, c; Supplementary Fig. 2c). No alteration of *Dnm1* and *Dnm3* transcripts was observed (Supplementary Fig. 2d). As expected, we observed a 2.9 fold increase in Ub-*Dnm2* due a splice switching from M-*Dnm2* to Ub-*Dnm2* by the removal of exon 12b (Fig. 2c), as removal of in-frame exon 12b leads to the production of Ub-*Dnm2* from the deleted allele (Fig. 2c, d). Therefore *Dnm2*ex12b$^{-/-}$ mice represent a splice switching model from M-*Dnm2* to Ub-*Dnm2*. This is only relevant in tissues where 12b is normally expressed and it should not affect expression in other tissues where it is not expressed.

*Dnm2*ex12b$^{-/-}$ female and male mice were phenotyped at 2 wks, 8 wks and 8 months (Fig. 2a). No significant difference in body weight nor in whole body strength (hanging test) was observed at any time-point analyzed (Fig. 2e; Supplementary Fig. 3a, b). At 2wks, corresponding to the main increase in M-*Dnm2* expression, there was no obvious difference between *Dnm2*ex12b$^{-/-}$ and wild-type mice in muscle weight and histology (Supplementary Fig. 3c, d). At 8 wks, corresponding to the end of postnatal muscle maturation and hypertrophy, *Dnm2*ex12b$^{-/-}$ mice performed similarly to wild-type mice for locomotor activity (actimetry and rotarod), muscle force (grip and hanging tests, in-situ maximal force), endurance (treadmill exhaustion test) and breathing capacity (plethysmography) (Supplementary Figs. 3a and 4a–c, Supplementary Table 2). No differences in TA muscle weight nor histology were observed neither at 8 wks of age (Supplementary Fig. 3c, d).

Next, a detailed phenotyping of *Dnm2*ex12b$^{-/-}$ mice was performed at 8 months for both genders. The maximal and specific force and the force-frequency relationship was not altered, the

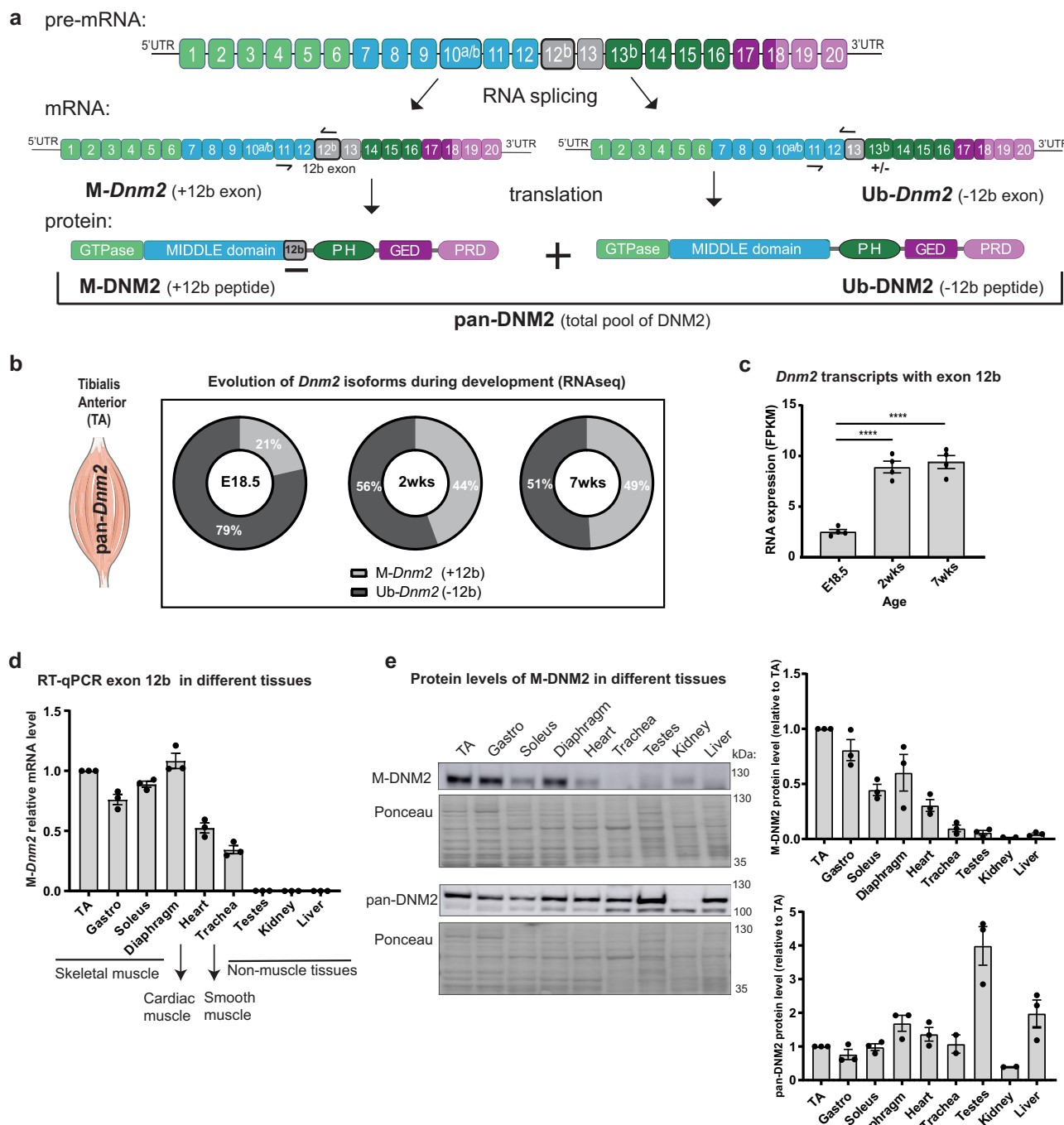

**Fig. 1 | Inclusion of exon 12b defines a DNM2 isoform only expressed in muscle tissue. a** Exonic and protein domain structure of DNM2 isoforms in muscle tissue. On the top, pre-mRNA containing all *Dnm2* exons. Exons 10a, 10b, 12b and 13b are alternatively spliced. *Dnm2* isoforms containing exon 12b are named as M-*Dnm2* (Muscle-specific *Dnm2*), while isoforms skipping exon 12b are referred as Ub-*Dnm2* (Ubiquitously expressed *Dnm2*). Exon 13b is not found in transcripts containing exon 12b. Ub-*Dnm2* isoforms can include or not exon 13b. The position of forward and reverse primers to detect both isoforms is indicated with black arrows in the mRNA. In the protein structure, 12b peptide is predicted to localize between PH (pleckstrin homology) domain and the stalk/middle domain. Pan-DNM2 englobes all DNM2 isoforms (M-DNM2 and Ub-DNM2). GED (GTPase effector domain), PRD (proline rich domain). **b** RNAseq data from wild-type skeletal muscle at different ages: E18.5 (embryonic day 18.5), 2 weeks and 7 weeks, showing proportions of *Dnm2* transcripts containing (M-*Dnm2*) or not (Ub-*Dnm2*) exon 12b (n = 4 mice/age).

The figure uses a vector from Servier Medical Art, provided by Servier, licensed under a Creative Commons Attributions 3.0 Unported License (http://smart.servier.com). **c** RNAseq data representing evolution in expression of M-*Dnm2* with the age. It is expressed as FPKM (Fragments Per Kilobase Million) (n = 4 mice/age, ****$P < 0.0001$ by one-way ANOVA with Tukey's post hoc test). **d** mRNA relative expression of M-*Dnm2* done by RT-qPCR in several muscle and non-muscle tissues from 5-week-old wild-type mice (n = 3 mice). Level of expression in different tissues was compared to expression in TA for each mouse. **e** Representative western blot with protein extracts from 5-week-old wild-type mouse tissues (n = 3 mice) probed with DNM2 pan-isoform antibody and M-DNM2 antibody (against 12b peptide). Right, fold increase of pan-DNM2 and M-DNM2 protein level compared to TA level in each mouse (n = 3 mice). **c–e** Data are represented as mean values ± SEM. Source data are provided as a Source Data file.

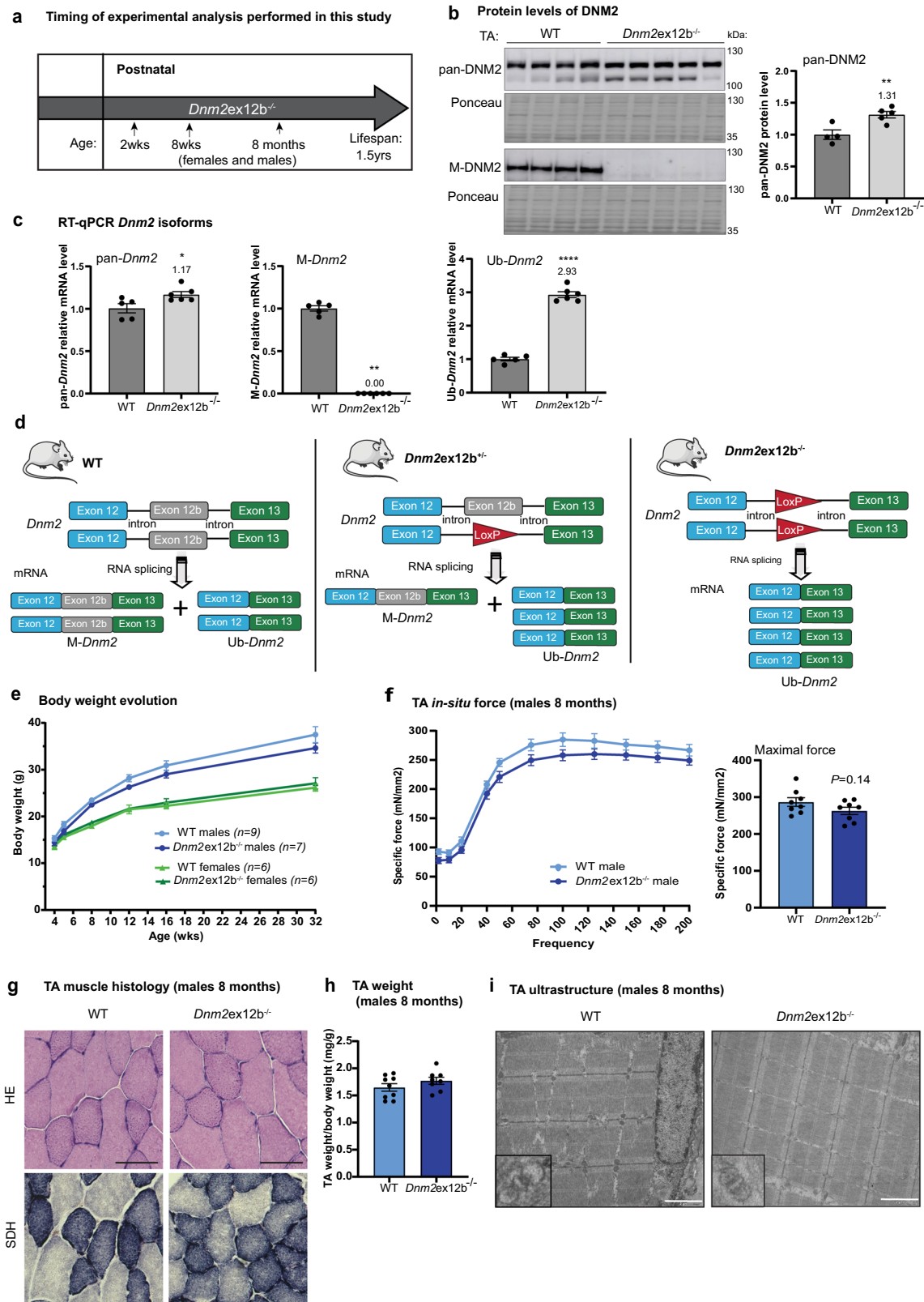

**a** Timing of experimental analysis performed in this study

**b** Protein levels of DNM2

**c** RT-qPCR *Dnm2* isoforms

**d**

**e** Body weight evolution

**f** TA *in-situ* force (males 8 months)

**g** TA muscle histology (males 8 months)

**h** TA weight (males 8 months)

**i** TA ultrastructure (males 8 months)

histology colorations did not display obvious anomalies nor CNM-like hallmarks, and analysis of the ultrastructure by electron microscopy revealed a normal organization of sarcomeres (Fig. 2f–i; Supplementary Fig. 5a, b). As DNM2, together with BIN1, are known to be main players in T-tubule formation and maintenance, we studied T-tubule orientation using potassium ferrocyanide staining and electron microscopy and we observed an increase in mis-oriented T-tubules in *Dnm2e*x12b$^{-/-}$ mice (Supplementary Fig. 5c). Immunofluorescence studies showed that DNM2 displayed similar localization in WT and *Dnm2e*x12b$^{-/-}$ mice (Supplementary Fig. 5d), as well as normal localization for DNM2 partners such as CHC and BIN1 (Supplementary Fig. 5e). Overall, these results indicate that the full removal of M-*Dnm2* is dispensable for muscle formation and maintenance.

**Fig. 2 | Constitutive deletion of exon 12b is dispensable for muscle formation and maintenance. a** Shema showing selected ages for phenotyping of the novel viable *Dnm2* mouse model with homozygous constitutive deletion of exon 12b. **b** Western blot with TA muscle protein extracts from wild-type (WT) and *Dnm2*ex12b$^{-/-}$ mice probed with pan-DNM2 and M-DNM2 antibody, (*n* = 5 mice/group, **$P$ = 0.0087 by two-tailed unpaired $t$ test) (**c**) *Dnm2* isoforms mRNA level by RT-qPCR in TA muscle (*n* = 5 mice/group, *$P$ = 0.0285 for pan-*Dnm2* comparison, and ****$P$ < 0.0001 for Ub-*Dnm2* comparison by two-tailed unpaired $t$ test. **$P$ = 0.0022 for M-*Dnm2* by two-tailed Mann–Whitney test). **d** Schema to illustrate the change in splicing in *Dnm2*ex12b mouse model (heterozygous and homozygous) after deletion of exon 12b (loxP sequence remaining instead of exon). Theoretical % of each isoform (*M-Dnm2*: *Ub-Dnm2*) is: 50:50 in wild-type (WT), 25:75 in *Dnm2*ex12b$^{+/-}$ and 0:100 in *Dnm2*ex12b$^{-/-}$. The figure uses a vector from Servier Medical Art, provided by Servier, licensed under a Creative Commons Attributions

3.0 Unported License (http://smart.servier.com). **e** Body weight evolution of WT and *Dnm2*ex12b$^{-/-}$ mice. Number of mice per group and ages of measurement are indicated in the figure. **f** Specific muscle force produced by TA muscle of WT and *Dnm2*ex12b$^{-/-}$ 8-month-old male mice when stimulated at different frequencies (Hertz, Hz). Right, specific maximal force is represented (*n* = 8 mice/group, two-tailed unpaired $t$ test). **g** Representative HE and SDH images of TA from male mice at 8 months of age. Scale bar = 50 μm. **h** TA weight ratio with body weight of 8-months-old WT (*n* = 9) and *Dnm2*ex12b$^{-/-}$ (*n* = 8) male mice (not significant (ns) by unpaired $t$ test). **i** Representative muscle image by electron microscopy, with magnification of a triad, in WT and *Dnm2*ex12b$^{-/-}$ muscles (*n* = 2 mice/group, females and males were analyzed with similar results). Scale bar = 2 μm. **b**, **c**, **e**, **f**, **h** Data are represented as mean values ± SEM. Source data are provided as a Source Data file.

## *Dnm2* splice switching towards Ub-*Dnm2* exacerbates the XLCNM phenotype

X-linked CNM (XLCNM) caused by *MTM1* loss-of-function mutations is one of the most severe CNM forms, characterized by early onset and profound hypotonia and muscle weakness accompanied by respiratory insufficiency. The *Mtm1*$^{-/y}$ mouse is a faithful model for XLCNM, that displays a CNM-like histology and develops a progressive muscle weakness and atrophy from 3 wks of age leading to premature death probably due to respiratory failure[41]. Heterozygous *Mtm1*$^{+/-}$ females are not known to be affected. Previously, we showed that reduction of pan-*Dnm2* to around 50% in all tissues efficiently rescues the lifespan and the different phenotypes of the *Mtm1*$^{-/y}$ mouse[16,37,38]. To increase the specificity of *Dnm2*-targeting to muscles, we targeted M-*Dnm2*. As it is tolerable to reduce M-*Dnm2* (see above), we generated *Mtm1*$^{-/y}$ *Dnm2*ex12b$^{-/-}$ and *Mtm1*$^{-/y}$ *Dnm2*ex12b$^{+/-}$ mice.

*Mtm1*$^{-/y}$ *Dnm2*ex12b$^{+/-}$ male mice were born at a lower ratio than expected based on Mendelian inheritance (5.0 to 5.2% compared to expected 25%), and no pups were obtained with the *Mtm1*$^{-/y}$ *Dnm2*ex12b$^{-/-}$ genotype (Fig. 3a; Supplementary Fig. 6a, b, Supplementary Tables 3, 4). The proportion of female *Mtm1*$^{+/-}$, *Mtm1*$^{+/-}$ *Dnm2*ex12b$^{+/-}$ and *Mtm1*$^{+/-}$ *Dnm2*ex12b$^{-/-}$ mice born didn't deviate significatively from the expected Mendelian ratio (Supplementary Fig. 6a, b, Supplementary Tables 3 and 4,). *Mtm1*$^{-/y}$ survived a median of 8 weeks (longest survivor: 17 wks), while lifespan for *Mtm1*$^{-/y}$ *Dnm2*ex12b$^{+/-}$ mice was dramatically reduced, with a median survival of 5 weeks (significantly decreased lifespan compared to *Mtm1*$^{-/y}$, $p$ = 0.0012) (Fig. 3b). Only 25% *Mtm1*$^{-/y}$ *Dnm2*ex12b$^{+/-}$ were alive at 5 wks and none reached 6 wks of age. Specific RT-qPCR revealed that M-*Dnm2* was decreased to 50% in *Mtm1*$^{-/y}$ *Dnm2*ex12b$^{+/-}$ compared to *Mtm1*$^{-/y}$ muscles, while Ub-*Dnm2* was increased by about 1.45 fold, validating the splice switching from M-*Dnm2* to Ub-*Dnm2* (Figs. 2d; 3c). Reduction of M-DNM2 was confirmed by western blot and a significant increase in pan-DNM2 protein levels was observed in *Mtm1*$^{-/y}$ mice as previously reported[37,38] with even higher increase in *Mtm1*$^{-/y}$ *Dnm2*ex12b$^{+/-}$ mice (Fig. 3d).

To explore further the effect of *Dnm2* splice switching towards Ub-*Dnm2* in *Mtm1*$^{-/y}$ mice, we performed motor, histological and ultrastructural analyses of different muscles at 4 wks of age. *Mtm1*$^{-/y}$ mice have a reduced body weight and motor function (hanging test) compared to wild-type controls, while the *Mtm1*$^{-/y}$ *Dnm2*ex12b$^{+/-}$ mice have a further reduction in body weight and cannot hang at all (Fig. 3e, f). *Mtm1*$^{-/y}$ *Dnm2*ex12b$^{+/-}$ mice appeared clearly more affected than *Mtm1*$^{-/y}$ mice as measured through a disease severity score assessing ptosis, kyphosis, breathing difficulties, position of hindlimbs when walking, ability to hang and difference of body weight (Fig. 3g, h; Supplementary Videos 1–3). The weight of different muscles was decreased in *Mtm1*$^{-/y}$ mice compared to wild-type and this difference was aggravated in *Mtm1*$^{-/y}$ *Dnm2*ex12b$^{+/-}$ mice (Supplementary Fig. 6c). Histological characterization showed worsening of the myofiber hypotrophy with no fibers larger than 30 μm in diameter (minFeret) in *Mtm1*$^{-/y}$ *Dnm2*ex12b$^{+/-}$

muscles compared to *Mtm1*$^{-/y}$ mice (Fig. 3I, j; Supplementary Fig. 6d, e). *Mtm1*$^{-/y}$ *Dnm2*$^{+/-}$ muscle fibers presented with a similar disruption to nuclei localization as *Mtm1*$^{-/y}$ (Fig. 3i, k), but with a higher percentage of fibers with mitochondrial disorganization, specifically aggregating in the periphery of the fibers (Fig. 3i, l). Diaphragm muscles from *Mtm1*$^{-/y}$ *Dnm2*ex12b$^{+/-}$ mice were smaller than *Mtm1*$^{-/y}$ mice, whilst nuclei position was similar across mice analyzed, with few internalized nuclei found in both genotypes (Supplementary Fig. 7a, b). Electron microscopy images identified partial misalignment of the Z-line, disorganization of sarcomeres and internalized nuclei in *Mtm1*$^{-/y}$ TA muscles, while *Mtm1*$^{-/y}$ *Dnm2*ex12b$^{+/-}$ mice displayed a stronger aggravation of the CNM phenotype with general disorganization in some fibers (Supplementary Fig. 7c). Regarding the phenotype in females, we observed that the total skipping of exon 12b induced a mild muscle phenotype in mice with a reduction in muscle force particularly at low frequencies as 2 Hz (twitch) (Supplementary Fig. 8c) and abnormal accumulation of oxidative staining in the center and periphery of muscle fibers (Supplementary Fig. 8e, f). However, there was no alteration of body and TA weights and hanging ability (Supplementary Fig. 8a, b, d).

Thus, in comparison to reduction of the total pool of DNM2 ('pan-*Dnm2*') which ameliorated the CNM phenotype in this mouse line[37], partial or complete *Dnm2* splice switching from the M-*Dnm2* to the Ub-*Dnm2* isoforms in the *Mtm1*$^{-/y}$ mouse was not therapeutic. Decreasing M-*Dnm2* and increasing Ub-*Dnm2* in a wild-type background was tolerated (*Dnm2*ex12b$^{-/-}$; see above) but significantly worsened the myopathic phenotypes of the *Mtm1*$^{-/y}$ male mouse and revealed a muscle phenotype in the *Mtm1*$^{-/y}$ female mouse. Altogether, these results may suggest that either the imbalance between the isoforms or the increase in Ub-*Dnm2* is a main cause of exacerbation of the phenotypes.

## Increase in Ub-DNM2 correlates with the XLCNM phenotype in mice

To assess if a specific *Dnm2* isoform is altered in the *Mtm1*$^{-/y}$ mouse, RT-qPCR and western blot were performed in TA muscles during the symptomatic phase at 5 wks of age. The levels of M-DNM2 were not altered during the disease state at both RNA and protein levels while pan-DNM2 proteins were significantly increased to 1.32 fold (Fig. 4). We conclude that the overall increase of DNM2 is due to a specific increase in the remaining pool of DNM2; Ub-DNM2 protein. The parallel decrease in Ub-*Dnm2* mRNA to 0.73 fold suggests a potential compensatory mechanism. Altogether, results presented here suggest the severity of CNM phenotypes of the *Mtm1*$^{-/y}$ mouse correlate with an increase of Ub-DNM2, but not with an alteration of M-DNM2.

## Differential impact of Ub-DNM2 and M-DNM2 on the CNM pathology

Based on the above data, we hypothesized that Ub-DNM2 is the main isoform linked to the CNM pathology in the *Mtm1*$^{-/y}$ mouse. To test this

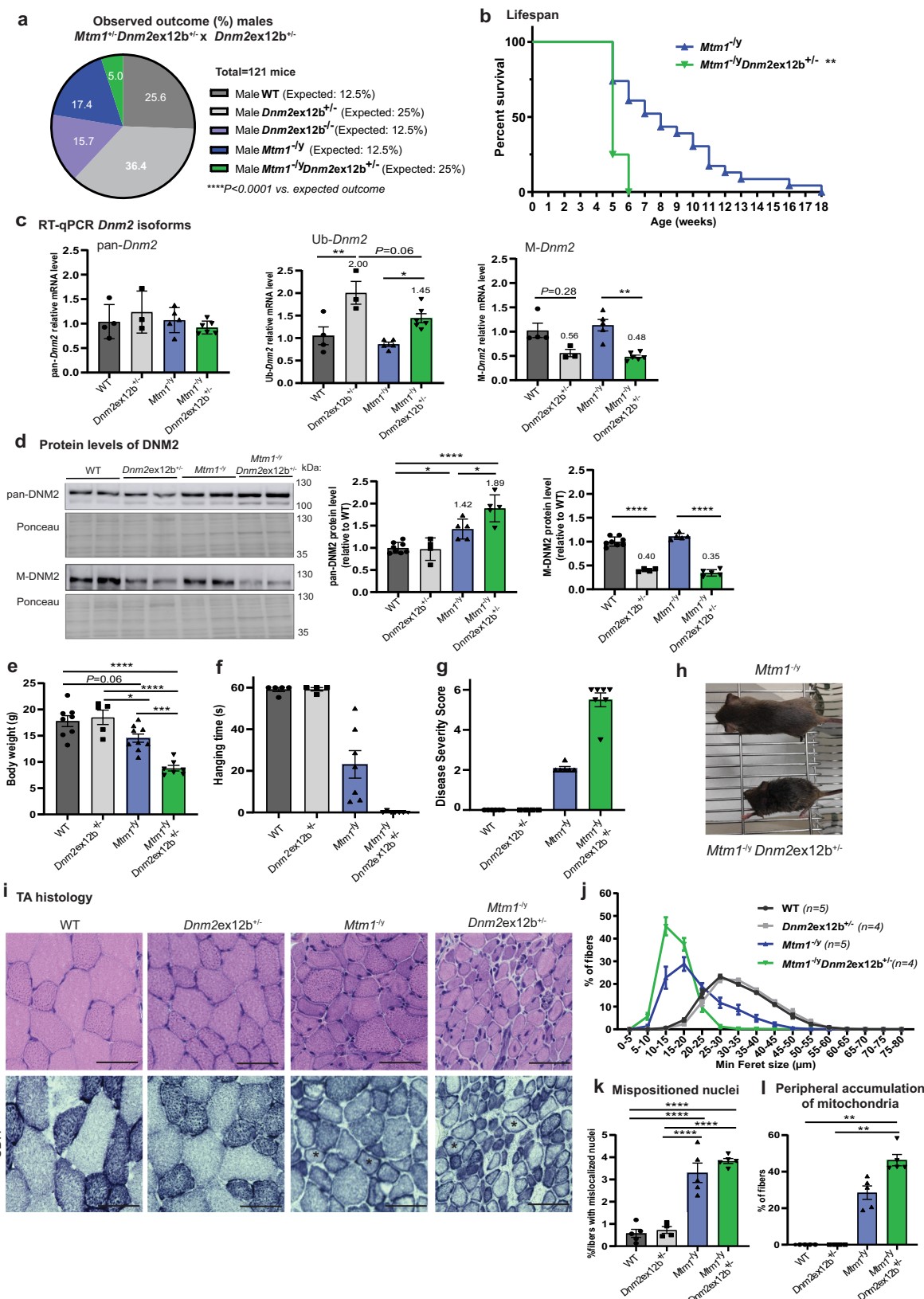

hypothesis in vivo, we overexpressed either Ub-DNM2 or M-DNM2 in wild-type mice and analyzed the resulting muscle phenotypes. Adeno-associated virus (AAV) expressing Ub-*DNM2* or M-*DNM2* were injected in the TA muscles of mice at 3 wks of age and muscles were analyzed 2 and 4 wks post-injection. Two weeks post-injection, both isoforms were overexpressed similarly at about 5–6 fold compared to the DNM2

endogenous level measured in control mice injected with empty AAV (Fig. 5a). Both isoforms decreased the specific muscle force and led to a significant muscle atrophy (Fig. 5b, Supplementary Fig. 9a, b). Classically muscle histology has been the main diagnostic tool to identify CNMs. Whilst centralized nuclei are present in most patients with severe CNM, some patients with *MTM1* or *DNM2* mutations presenting

**Fig. 3 | *Dnm2* splice switching towards ubiquitous isoforms is worsening XLCNM muscle phenotype and lifespan. a** Observed and expected birth ratio (in %). No pups were observed with *Mtm1*$^{-/y}$ *Dnm2*ex12b$^{-/-}$ genotype (expected: 12.5%). Observed ratio does not follow Mendelian inheritance (****$P < 0.0001$, two-tailed chi-square test). **b** Percentage of survival of *Mtm1*$^{-/y}$ *Dnm2*ex12b$^{+/-}$ ($n = 8$) and *Mtm1*$^{-/y}$ ($n = 23$) mice (**$P = 0.0012$, Log-rank (Mantel−Cox) test). **c** RT-qPCR of *Dnm2* isoforms in TA from WT ($n = 4$), *Dnm2*ex12b$^{+/-}$ ($n = 3$), *Mtm1*$^{-/y}$ ($n = 5$), *Mtm1*$^{-/y}$ *Dnm2*ex12b$^{+/-}$ ($n = 6$) mice/group (Ub-*Dnm2* **$P = 0.0031$ and *$P = 0.0225$ and M-*Dnm2* **$P = 0.0046$ by one-way ANOVA with Sidak's post hoc test). **d** Protein levels of pan-DNM2 and M-DNM2 in TA extracts from WT ($n = 8$), *Dnm2*ex12b$^{+/-}$ ($n = 4$), *Mtm1*$^{-/y}$ ($n = 5$), *Mtm1*$^{-/y}$ *Dnm2*ex12b$^{+/-}$ ($n = 5$) mice/group. Pan-DNM2 comparison (from left to right *$P = 0.0133$, *$P = 0.0142$, ****$P < 0.0001$) and M-DNM2 comparison (****$P < 0.0001$) by one-way ANOVA with Sidak's post hoc test). **e** Body weight of WT ($n = 8$), *Dnm2*ex12b$^{+/-}$ ($n = 5$), *Mtm1*$^{-/y}$ ($n = 9$), *Mtm1*$^{-/y}$ *Dnm2*ex12b$^{+/-}$ ($n = 7$) mice (****$P < 0.0001$, *$P = 0.0395$, ***$P = 0.0006$ by one-way ANOVA with Tukey's post hoc test). **f** Hanging time for a maximum of 60 s (each dot represents the average of

three repetitions), and (**g**) Disease Severity Score (DSS) ranging from 0 to 6 (the higher is the DSS, more severe is the phenotype) measured for WT ($n = 5$), *Dnm2*ex12b$^{+/-}$ ($n = 4$), *Mtm1*$^{-/y}$ ($n = 7$), *Mtm1*$^{-/y}$ *Dnm2*ex12b$^{+/-}$ ($n = 7$) mice/group. **h** Representative image of *Mtm1*$^{-/y}$ *Dnm2*ex12b$^{+/-}$ and its littermate *Mtm1*$^{-/y}$ mouse. **i** Representative HE and SDH images of TA sections. Scale bar = 50 μm. Representative fibers with aggregation of mitochondria in the periphery were indicated with an asterisk (*).**j** Minimum Feret diameter (Min Feret) of TA fibers grouped into 5 μm intervals, including fibers with size bigger than lower limit and smaller or equal than upper limit. **k** Percentage of fibers with abnormal (centralized and internalized) nuclei position (****$P < 0.0001$ by one-way ANOVA with Tukey's post hoc test), and (**l**) Percentage of fibers with abnormal SDH staining as * fiber in (**i**) ($n = 5$ mice/group, except $n = 4$ for *Dnm2*ex12b; **$P = 0.0047$ by Kruskal-Wallis test with Dunn's post hoc test). **c**−**l** All analysis corresponds to 4-week-old male mice. **c**−**g**, **j**−**l** Data are represented as mean values ± SEM. Source data are provided as a Source Data file.

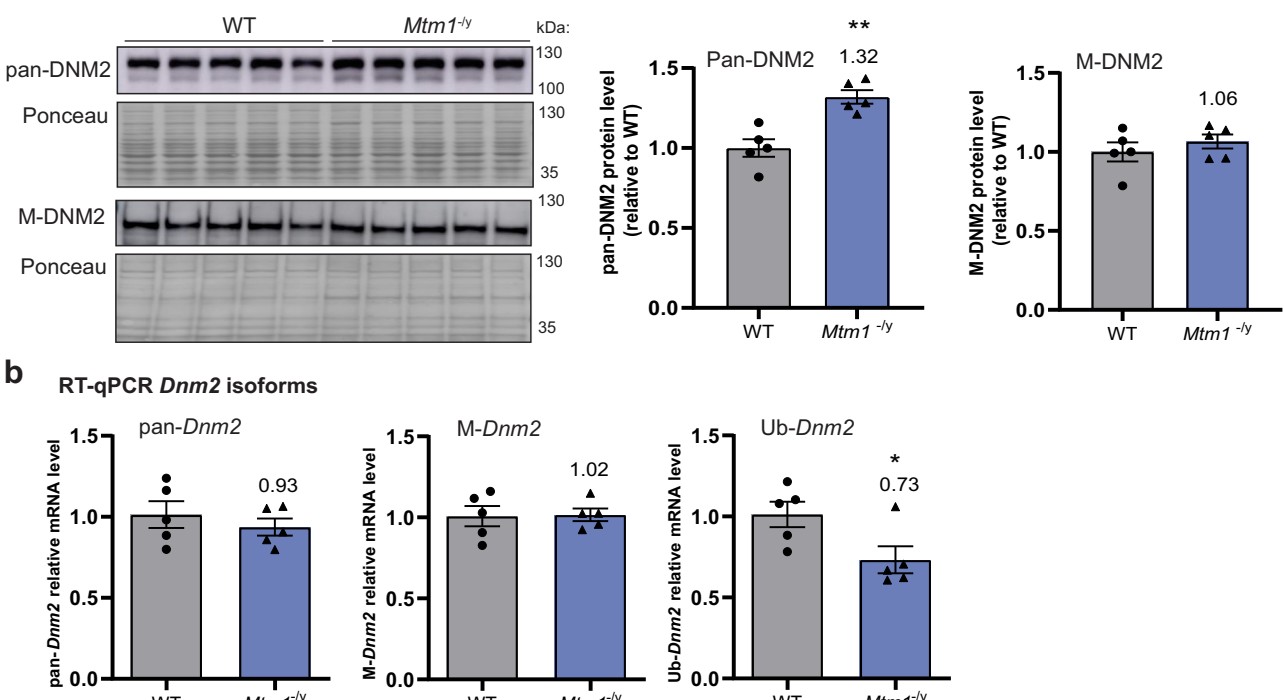

**a** **Protein level of DNM2 isoforms**

**b** **RT-qPCR *Dnm2* isoforms**

**Fig. 4 | Increased pan-DNM2 protein levels in XLCNM do not correlate with increased level of M-DNM2. a** Western blot with TA muscle protein extracts probed with pan-DNM2 and M-DNM2 antibodies in sick 5-week-old *Mtm1*$^{-/y}$ mice. Densitometer of the blot relative to WT is showed at the right ($n = 5$ mice/group, **$P = 0.0018$ by two-tailed unpaired $t$ test). M-DNM2 was not found significantly

increased in *Mtm1*$^{-/y}$ mice. **b** mRNA level of *Dnm2* isoforms in sick 5-week-old *Mtm1*$^{-/y}$ mice, RNA was extracted from same mice of figure (**a**) ($n = 5$ mice/group, *$P = 0.0317$ by two-tailed Mann–Whitney $t$ test). **a**, **b** Data are represented as mean values ± SEM. Source data are provided as a Source Data file.

with a mild adult form display muscle biopsies with necklace fibers characterized by the apparition of a basophilic subsarcolemmal ring stained by oxidative reactions[42–44]. Overexpression of both *DNM2* isoforms correlated with myofiber hypotrophy and abnormal accumulation of oxidative NADH-TR and SDH activities supporting mislocalization of reticulum and mitochondria (Fig. 5c, d; Supplementary Fig. 9c, d). Ub-DNM2 overexpression correlated with a high number of mis-localized nuclei, mainly found at the center of myofibers (Fig. 5d–f). Electron microscopy confirmed the presence of mislocalized nuclei surrounded by reticulum and organelles rather than sarcomeres (Supplementary Fig. 9e). The number of fibers with mislocalized nuclei was much lower with M-DNM2 expression. However, M-DNM2 overexpression was linked to the presence of a high number of necklace fibers, where these necklaces are positive for the oxidative

NADH-TR and SDH activities and for periodic acid Schiff (PAS staining) and thus composed of reticulum, mitochondria, and glycogen (Fig. 5d, g). Conversely, Ub-DNM2 overexpression was not associated with the presence of necklaces. Electron microscopy confirmed the necklaces are a collapse of glycogen and reticulum, and immunofluorescence labelling with the TOMM20 outer mitochondria membrane protein validated the presence of mitochondria (Fig. 5h; Supplementary Fig. 9e). In addition, dysferlin, a protein implicated in membrane remodeling and muscle repair, was present in necklaces while actin was excluded (Fig. 5i). Similar results were obtained 4 wks postinjection (Supplementary Fig. 10). As overexpression of different DNM2 isoforms in WT mice leads to variable CNM pathology, we aimed to investigate if exogenous expression of DNM2 isoforms would aggravate *Mtm1*$^{-/y}$ muscle phenotypes. AAV expressing Ub-*DNM2* or M-

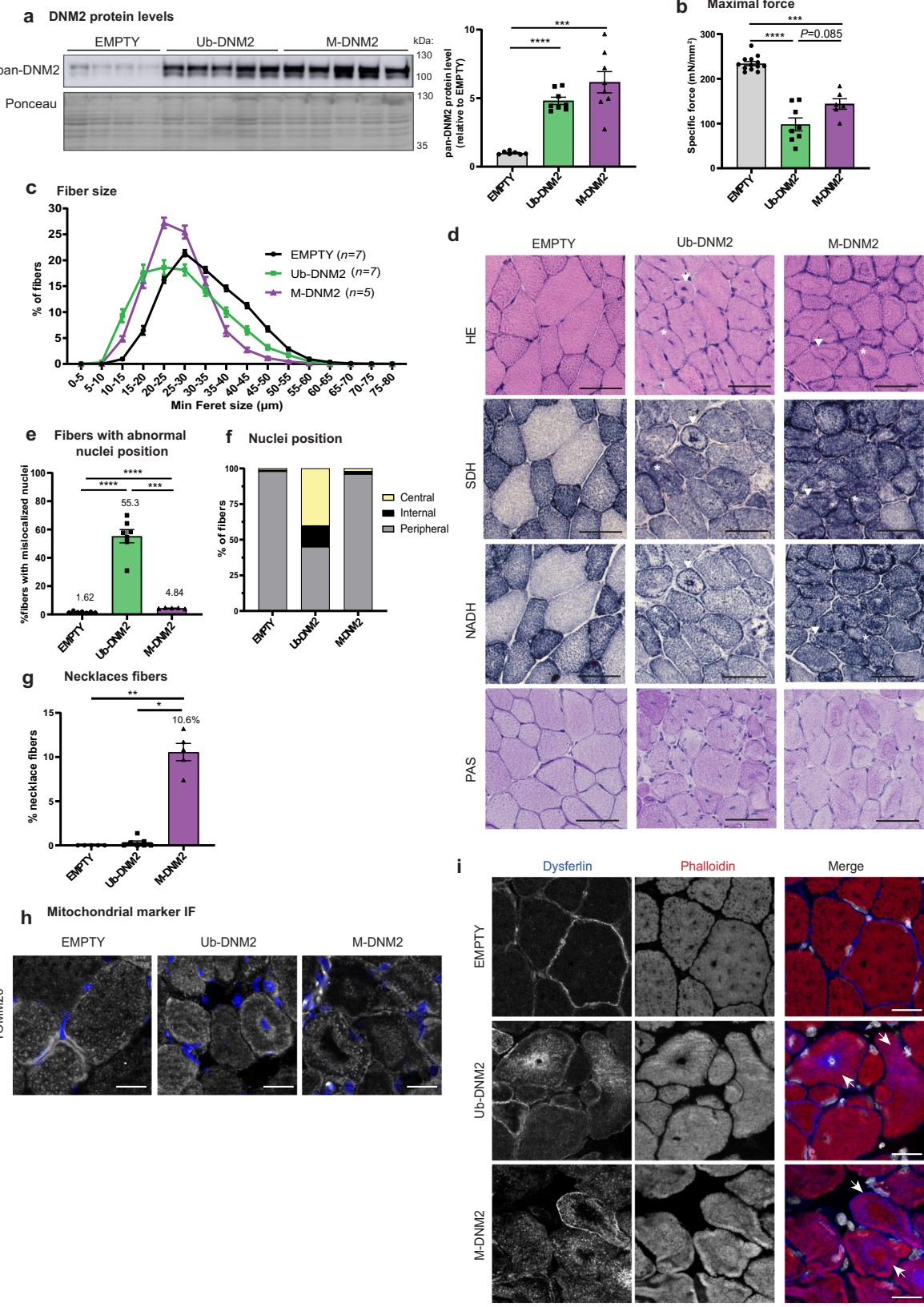

*DNM2* were injected in TA from 3-week-old *Mtm1*[−/y] mice and resulted in a slight and similar aggravation of *Mtm1*[−/y] histological phenotypes 2 weeks later (Supplementary Fig. 11). In conclusion, overexpression in WT muscle of the Ub-DNM2 correlates with histological hallmarks of severe neonatal CNM while expression of the M-DNM2 is linked to hallmarks found in some mild adult CNM cases.

## Overexpressed M-DNM2 but not Ub-DNM2 alters BIN1 and clathrin localization

As Ub-DNM2 and M-DNM2 expression induces different CNM phenotypes in WT mice, we next investigated their subcellular localization. Endogenous DNM2 is localized at the I-band encompassing the Z-line[27]. While both isoforms localized around the Z-line in AAV−mediated

**Fig. 5 | Overexpression of Ub-DNM2 and M-DNM2 correlate with different CNM-like phenotypes 2 weeks post-injection. a** Representative western blot and DNM2 protein level quantification in TA revealed by pan-DNM2 antibody compared to endogenous DNM2 in AAV-empty (Empty ($n = 7$), Ub-DNM2 and M-DNM2 ($n = 8$) mice/group; ****$P < 0.0001$ and ***$P = 0.0007$ by Brown-Forsythe and Welch ANOVA with Dunnett's T3 post hoc test). **b** Maximal specific force of TA muscle (Empty ($n = 13$), Ub-DNM2 ($n = 8$), M-DNM2 ($n = 6$); ****$P < 0.0001$ and ***$P = 0.0006$ by Brown-Forsythe and Welch ANOVA with Dunnett's T3 post hoc test). **c** Minimum Feret diameter of TA fibers from AAV-transduced muscles is represented grouped into 5 μm intervals. **d** Representative sections from AAV-transduced muscle stained with HE, SDH, NADH and PAS staining. Arrows point a fiber with central and peripheral collapse of mitochondria for Ub-DNM2 condition and, a fiber with a necklace for M-DNM2 muscle. Asterisk (*) points a fiber with central nuclei for Ub-DNM2 and, a necklace collapse in the center of the fiber for M-DNM2. Scale bar = 50 μm. **e** Percentage of myofibers with abnormal nuclei position (internal and central)

(****$P < 0.0001$ and ***$P = 0.0002$ by Brown-Forsythe and Welch ANOVA with Dunnett's T3 post hoc test) and (**f**) those fibers classified by nuclei position: peripheral (nuclei in contact with sarcolemma), central (at least one nucleus in the center of the myofiber) and internal (nucleus not touching the sarcolemma without being positioned in the center). Empty and Ub-DNM2 ($n = 7$) and M-DNM2 ($n = 5$), mice/group analyzed. **g** Percentage of fibers with necklace as the fiber pointed by an arrow for M-DNM2 in (**d**) (Empty and M-DNM2 ($n = 5$) and Ub-DNM2 ($n = 7$) mice/group, **$P = 0.0021$ and *$P = 0.0196$ by Kruskal-Wallis test with Dunn's post hoc test). **h** Representative TA transversal sections stained by immunofluorescence using TOMM20 antibody with nuclei shown in blue and TOMM20 in grey or (**i**) stained by dysferlin antibody and phalloidin fluorescent probe (binds to filamentous actin) with nuclei shown in grey. Images displayed are projections of confocal stacks. Scale bar = 20 μm. Stainings were repeated with similar result in 2 independent experiments with different mouse sample. **a–c, e, g** Data are represented as mean values ± SEM. Source data are provided as a Source Data file.

overexpressing muscles as observed for endogenous DNM2, Ub-DNM2 was also concentrated around centralized nuclei, while M-DNM2 was observed as a punctate pattern at the sub-sarcolemma (Fig. 6a, b). M-DNM2 positive longitudinal structures were also observed in myofibers. To investigate if the specific localization of M-DNM2 alters the localization of other proteins, immunofluorescence for DNM2 physical and functional partners was performed. BIN1, and to a lesser extent clathrin heavy chain (CHC) mislocalized on the longitudinal M-DNM2 puncta (Fig. 6c, d). In contrast BIN1 and CHC were found on T-tubules and at the Z-line, respectively, under normal conditions (empty) and when Ub-DNM2 was overexpressed. In addition to the specific recruitment of BIN1 and CHC to M-DNM2, we found a strong increase in BIN1 and CHC protein levels by western blot (Fig. 6e). This increase was found with overexpression of both Ub-DNM2 and M-DNM2 isoforms. Thus, increased level of pan-DNM2 triggers an increase in BIN1 and CHC levels, while only overexpressed M-DNM2 mislocalizes BIN1 and CHC. We checked if the trapping of BIN1 affects T-tubule structure, and observed the localization of DHPR, the T-tubule voltage-sensing receptor, was altered, more diffuse and longitudinal, compared to the normal transversal orientation (Supplementary Fig. 12a). Abnormal orientation of T-tubules was confirmed through ferrocyanate staining (Supplementary Fig. 12b). There was also an alteration of T-tubules orientation with overexpression of Ub-DNM2, albeit to a lesser extent.

To check whether the ability of M-DNM2 to localize as punctate structures is either an intrinsic property of M-DNM2 or linked to binding of a muscle-specific structure, DNM2 isoforms were exogenously expressed in non-muscle (COS) cells followed by vesicular fixation to remove cytosolic DNM2. Both isoforms colocalized with CHC, but M-DNM2 vesicular structures were enlarged compared to Ub-DNM2, reminiscent of the longitudinal subsarcolemmal punctate staining for M-DNM2 and CHC observed in myofibers (Supplementary Fig. 12c). These in vivo and in cellulo data suggest that M-DNM2 forms more higher-order structures compared to Ub-DNM2.

## M-DNM2 forms larger oligomers more sensitive to GTP-induced depolymerization

In order to better define the biochemical differences between Ub-DNM2 and M-DNM2 that can potentially underlie the differential in vivo impact of both isoforms, we characterized the GTPase activity and oligomerization properties of recombinant DNM2 isoforms. Human Ub-DNM2 and M-DNM2 proteins were produced with the Baculovirus system in Sf9 insect cells, purified with a resin coated with the SH3 domain of BIN1, and analyzed through gel filtration. Both DNM2 isoforms were efficiently purified through binding to BIN1 SH3 domain to a similar degree (Supplementary Fig. 13a). Gel filtration revealed M-DNM2 showed a higher hydrodynamic radius compared to Ub-DNM2, as identified by the shift of the curve to the left (Fig. 7a). To assess if this shift in gel filtration corresponds to higher-order assembly of M-DNM2, both DNM2 isoforms were analyzed by size-exclusion

chromatography combined with multiangle light scattering (SEC-MALS). SEC elution profile showed again Ub-DNM2 is eluting later than M-DNM2 (12.5 ml vs. 11.5 ml) and MALS confirmed this is due to the presence of larger oligomers of M-DNM2 with average molecular mass from 400 to 470KDa while Ub-DNM2 formed oligomer species from 200 to 350KDa (Supplementary Fig. 13b).

Increasing NaCl concentrations triggers dynamin depolymerization[45]. Sedimentation assays with NaCl concentrations ranging from 25 mM to 300 mM showed that M-DNM2 was present at higher oligomer states compared to Ub-DNM2 at higher NaCl concentrations (Fig. 7b, Supplementary Fig. 13c). GTP binding promotes dynamin disassembly while GMP-PCP (PCP) is a non-hydrolysable analog of GTP that triggers dynamin self-assembly without requiring the presence of lipids. Sedimentation assays with GMP-PCP showed that M-DNM2 formed larger oligomers compared to Ub-DNM2, suggesting M-DNM2 has a much higher propensity to self-assemble (Fig. 7c, Supplementary Fig. 14a). Next, GTP was used to assess the stability of oligomers at low salt concentrations of 37.5 mM NaCl. In this condition, both isoforms formed oligomers as expected, and M-DNM2 was more sensitive to GTP-induced depolymerization (Fig. 7d, Supplementary Fig. 14b).

To assess more specifically the difference in organization of the DNM2 isoform oligomers, electron microscopy and negative staining were performed. Organized ring and horseshoe structures were observed predominantly with M-DNM2 and only rarely with Ub-DNM2 (Fig. 7e), even at high ionic strengths (Supplementary Fig. 14c). Addition of GMP-PCP promoted the formation of spiral structures only for M-DNM2 but not for Ub-DNM2. These spirals appear similar to dynamin spirals formed around the neck of vesicles budding from the plasma membrane during endocytosis[46]. Overall, compared to Ub-DNM2, M-DNM2 forms higher oligomer structures that are more resistant to salt-induced depolymerization. To investigate if the difference in oligomerization between M-DNM2 and Ub-DNM2 correlates with a difference in GTPase activity, GTPase activity was measured with increasing NaCl concentrations. Both isoforms displayed similar basal GTPase activity at incremental salt concentrations (Fig. 7f). However, in presence of lipids, M-DNM2 displayed a significantly higher GTPase activity at ≥110 mM salt concentration (Fig. 7g). In addition, M-DNM2 presented faster GTP hydrolysis as well as faster GTP-induced depolymerization compared with Ub-DNM2 (Fig. 7h, i, Supplementary Fig. 15a, b). Overall, Ub-DNM2 and M-DNM2 isoforms differ on their oligomerization, enzymatic activity and sensitivity to GTP-induced depolymerization.

To assess membrane fission activity of both DNM2 isoforms in a cellular context, we performed an ex vivo tubulation assay as previously described[47,48]. BIN1 expression in COS cells induced tubule-like membrane structures originating from the plasma membrane that can recruit exogenous DNM2. Increasing the ratio of DNM2 promotes severing of BIN1 tubules into smaller tubules or vesicles. BIN1 and

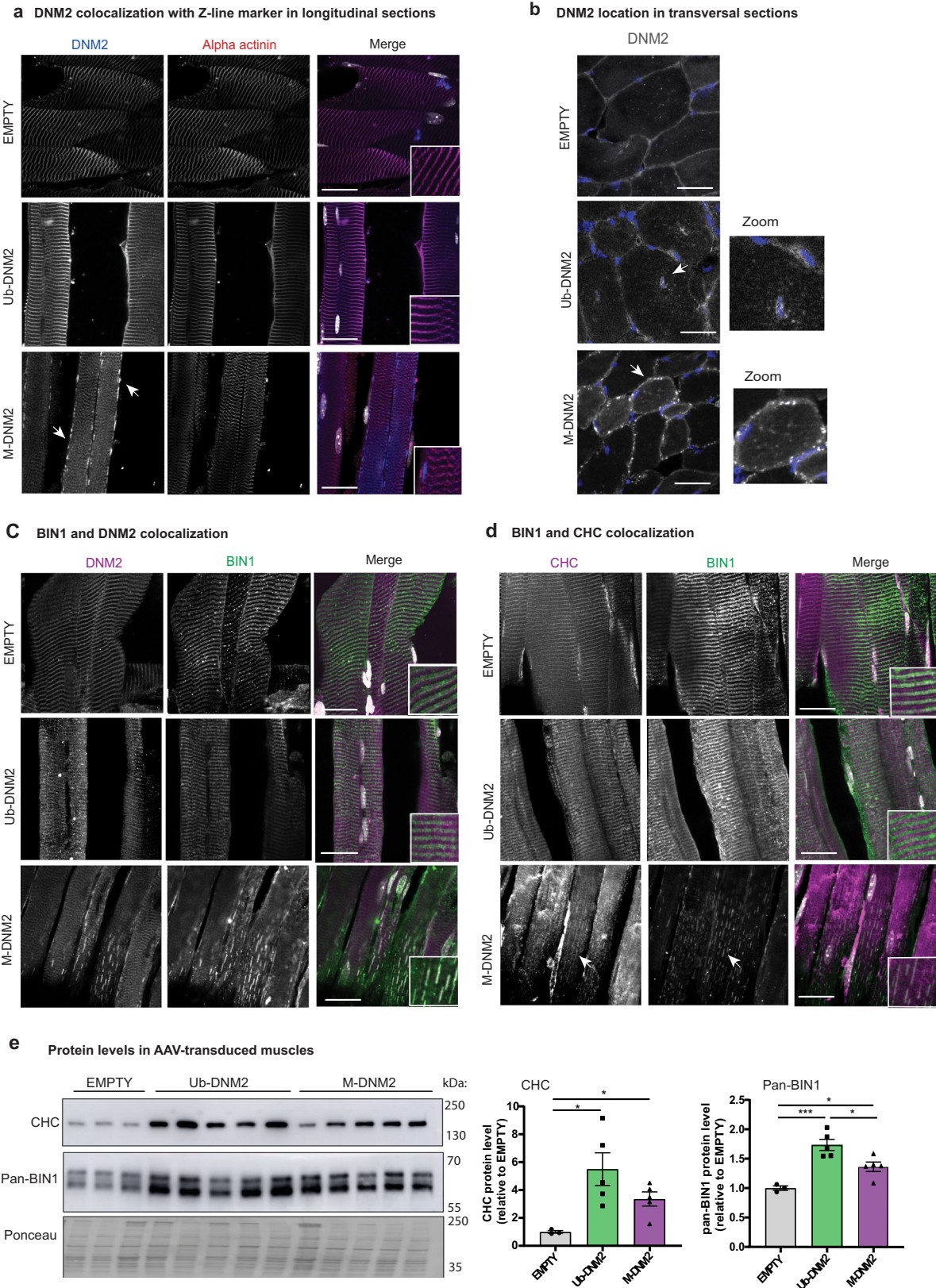

**a** DNM2 colocalization with Z-line marker in longitudinal sections

**b** DNM2 location in transversal sections

**c** BIN1 and DNM2 colocalization

**d** BIN1 and CHC colocalization

**e** Protein levels in AAV-transduced muscles

DNM2 colocalize on the tubules and the vesicles[47,48]. We observed M-DNM2 expression correlated with a slightly higher proportion of cells with vesicles or short tubules compared to Ub-DNM2 (Fig. 8a). Of note, a CNM mutation (DNM2-S619L) in the Ub-DNM2 isoform strongly increased tubules fission by ~30% (Fig. 8b). We thus investigated if the addition of the S619L mutation had the same impact in a M-DNM2 isoform context and observed only a slight increase in the proportion of cells with vesicular pattern (~3%) compared to its equivalent WT form (Fig. 8b). In conclusion, M-DNM2 forms larger oligomers that are more sensitive to GTP-induced depolymerization, while Ub-DNM2 oligomers are resistant to depolymerization and more sensitive to the CNM mutation.

**Fig. 6 | M-DNM2 overexpression, but not Ub-DNM2, alters BIN1 and clathrin localization. a** TA longitudinal sections of AAV transduced muscles were stained by immunofluorescence using DNM2 and alpha actinin (marker of Z-disk) antibodies. Merge of both signals is shown at the right with nuclei in grey. **b** TA transversal sections of AAV transduced muscles were stained by immunofluorescence using DNM2 antibody. Nuclei is shown in blue and DNM2 signal in grey. Arrows and zoom (x2) indicate DNM2 signal accumulation around centralize nuclei with Ub-DNM2 overexpression, while M-DNM2 overexpression shows punctuate subsarcolemmal signal. **c** TA longitudinal sections of AAV transduced muscles were stained by immunofluorescence using DNM2 and BIN1 antibodies and (**d**) CHC and BIN1 antibodies. Merge of both signals is shown at the right with nuclei in grey. The arrow indicates colocalization of BIN1 and CHC in longitudinal. **e** Western blot using

CHC and pan-BIN1 antibodies in AAV transduced muscles and quantification of protein levels compared to control condition (EMPTY) at the right (EMPTY (n = 3), Ub-DNNM2 and M-DNM2 (n = 5) mice/group; CHC analysis: Empty vs. Ub-DNM2 (*P = 0.0388) and vs. M-DNM2 (*P = 0.0190) by Brown-Forsythe and Welch ANOVA with Dunnett's T3 post hoc test and BIN1 analysis: Empty vs. Ub-DNM2 (***P = 0.0006 and Empty) and vs. M-DNM2 (*P = 0.0458) and M-DNM2 vs. Ub-DNM2 (*P = 0.0200) by one-way ANOVA with Tukey's post hoc test. Data are represented as mean values ± SEM. Source data are provided as a Source Data file. **a**–**d** Images displayed are projections of confocal stacks and scale bar = 20 μm. Stainings were repeated with the same result in 2 independent experiments with muscle sections from different mice.

## Discussion

In this study, we investigated the molecular regulation and physiological relevance of DNM2 and its involvement in the CNM pathology. In particular, we characterized DNM2 isoforms in skeletal muscle that encompass the ubiquitous isoform (Ub-DNM2) together with a muscle-specific isoform (M-DNM2) containing a peptide encoded by alternative exon 12b. Modulation of M-DNM2 level showed M-DNM2 is dispensable for muscle maturation and correlates with a mild adult form of CNM when overexpressed. In addition, increase of the Ub-DNM2 isoform through genetic splice switching exacerbated the phenotypes of $Mtm1^{-/y}$ (XLCNM mouse), and correlated with a severe CNM form when overexpressed. In agreement, Ub-DNM2 but not M-DNM2 was found increased in the $Mtm1^{-/y}$ mouse. Comparison of the behavior and function of Ub-DNM2 and M-DNM2 in vivo, in cells and in vitro revealed Ub-DNM2 membrane fission activity is more sensitive to a main CNM mutation and more resistant to GTP-induced depolymerization than M-DNM2, and Ub-DNM2 has more impact on the pathology in mouse muscles (Fig. 9).

Apparition of M-DNM2 correlates with late muscle maturation and muscle hypertrophy, while Ub-DNM2 is found in muscle from the myoblast stage and in most other tissues. The present data suggests these isoforms have a different function in muscle. First, increasing the Ub-DNM2 level in WT mice through AAV-mediated expression creates a severe CNM phenotype, while increasing the M-DNM2 correlates with a mild CNM phenotype that is similar to a subset of adult onset CNM patients characterized by necklace fibers (Fig. 5). Secondly, removing M-DNM2 (exon 12b KO mouse) did not lead to a CNM or major muscle defects, while removing all DNM2 isoforms specifically in skeletal muscle fosters muscle maturation defects with reduction in muscle mass, structural alteration of myofibers, and with a metabolic imbalance that results in a reduced lifespan of 10–14 days[30]. Thirdly, Ub-DNM2 is upregulated in $Mtm1^{-/y}$ mice, a faithful model of XLCNM, while M-DNM2 levels do not change. Fourth, the KO of exon 12b creates a splice switching from the M-$Dnm2$ to the Ub-$Dnm2$ isoform that is worsening the CNM phenotypes of the $Mtm1^{-/y}$ mice (Fig. 3). These data support Ub-DNM2 as the main isoform implicated in the pathology of XLCNM. In the absence of MTM1, Ub-DNM2 is increased and further increase of Ub-DNM2 through splice switching or AAV-mediated overexpression aggravates the phenotypes linked to MTM1 loss (Fig. 3 and Supplementary Fig. 11). This suggests the level of Ub-DNM2 is a modifier of XLCNM severity in males. In addition, $Dnm2$ isoform splice switching in $Mtm1^{+/-}$ females revealed a mild CNM phenotypes not seen in $Mtm1^{+/-}$ females (Supplementary Fig. 8). Of note, female XLCNM patients display a wide spectrum of clinical severity ranging from severe neonatal to asymptomatic that does not correlate with X-inactivation[43,49].

To decipher the differences in molecular and cellular functions of both DNM2 isoforms, we performed experiments in vitro, in cultured cells and in vivo in AAV-transduced myofibers. M-DNM2 formed larger oligomers as seen by gel filtration, SEC-MALS, sedimentation assay and negative staining. Noteworthy, M-DNM2 self-assembled into numerous rings and large spirals at physiological salt concentration (Fig. 7).

In accordance, M-DNM2 formed larger vesicle-like structure in cultured cells and puncta at the sarcolemma of myofibers compared to Ub-DNM2 (Fig. 6, Supplementary Fig. 11). These M-DNM2 structures recruited both CHC and BIN1, key regulators of endocytosis, costameres and T-tubules. In these myofibers, BIN1 was mislocalized away from its usual transversal localization, leading alteration of T-tubule orientation and organization. In contrast, we observed an increased turnover rate of M-DNM2 oligomers in presence of GTP compared to Ub-DNM2 oligomers. Finally, M-DNM2 was slightly more efficient than Ub-DNM2 to fission membrane tubules induced by BIN1 overexpression in cells (Fig. 8). In addition, the GTPase activity of M-DNM2 appeared more activated in the presence of lipids (Fig. 7). The muscle-specific peptide encoded by exon 12b is located on the dynamin structure near the auto-inhibitory stalk-PH domains interface which is released upon binding of the PH domain to membrane[5,17,50]. We hypothesize the presence of this muscle-specific peptide in M-DNM2 promotes either the opening of the auto-inhibitory interface or the stability of the open conformation, leading to increased oligomer state of DNM2 as noted in our in vitro assays (Fig. 7).

CNM mutants introduced in the Ub-DNM2 isoform have a higher propensity to self-assemble and are resistant to GTP-induce depolymerization[26,51–53]. This is especially the case of the R369W, R465W or S619L CNM mutants. Moreover, we found the addition of the CNM mutation S619L in Ub-DNM2 exacerbates its membrane fission properties in cells compared to the impact in M-DNM2, which may be already in a pre-activated conformation. Altogether, these molecular data support the Ub-DNM2 isoform is the main contributors of CNM pathogenesis as it is more sensitive to the CNM mutation, and this is in agreement with our in vivo data showing Ub-DNM2 overexpression in a WT and CNM ($Mtm1^{-/y}$) context has a more pathological impact than M-DNM2. Conversely, while M-DNM2 forms higher-order oligomers it has a higher GTP-induced depolymerization and is less sensitive to the CNM mutant in the membrane fission assays. The fact that M-DNM2 has an opposite sensitivity to GTP-induced depolymerization than the CNM mutants, compared to Ub-DNM2, is most probably at the basis of its reduced toxicity in vivo (Fig. 9).

In addition, BIN1-CNM mutants leading to truncation of the SH3 domain cause loss of DNM2 binding[54]. Also, DNM2-CNM mutants are more resistant to BIN1-mediated inhibition and increase fission of BIN1-membrane tubules in a Ub-DNM2 context[51]. Thus, CNM mutations in DNM2 disrupt the Stalk-PH interface and auto-inhibition while BIN1 mutants disrupt BIN1-mediated inhibition of DNM2. Noteworthy, BIN1 is inhibiting the GTPase activity of Ub-DNM2 but not of M-DNM2[17].

Here, we also investigated if targeting specifically M-DNM2 through removal of exon 12b is efficient to rescue the CNM hallmarks of the $Mtm1^{-/y}$ mouse, to potentially develop a therapy targeting specifically skeletal muscle. However, genetic removal of exon 12b leading to splice switching from M-$Dnm2$ to Ub-$Dnm2$ while maintaining a normal pan-$Dnm2$ level aggravated the $Mtm1^{-/y}$ phenotypes (Fig. 3). Targeting of pan-DNM2 could be achieved by RNA interference mechanism mediated by either ASO, AAV-shRNA or siRNA complementary to $Dnm2$ ubiquitous exons and leading to reduction in its

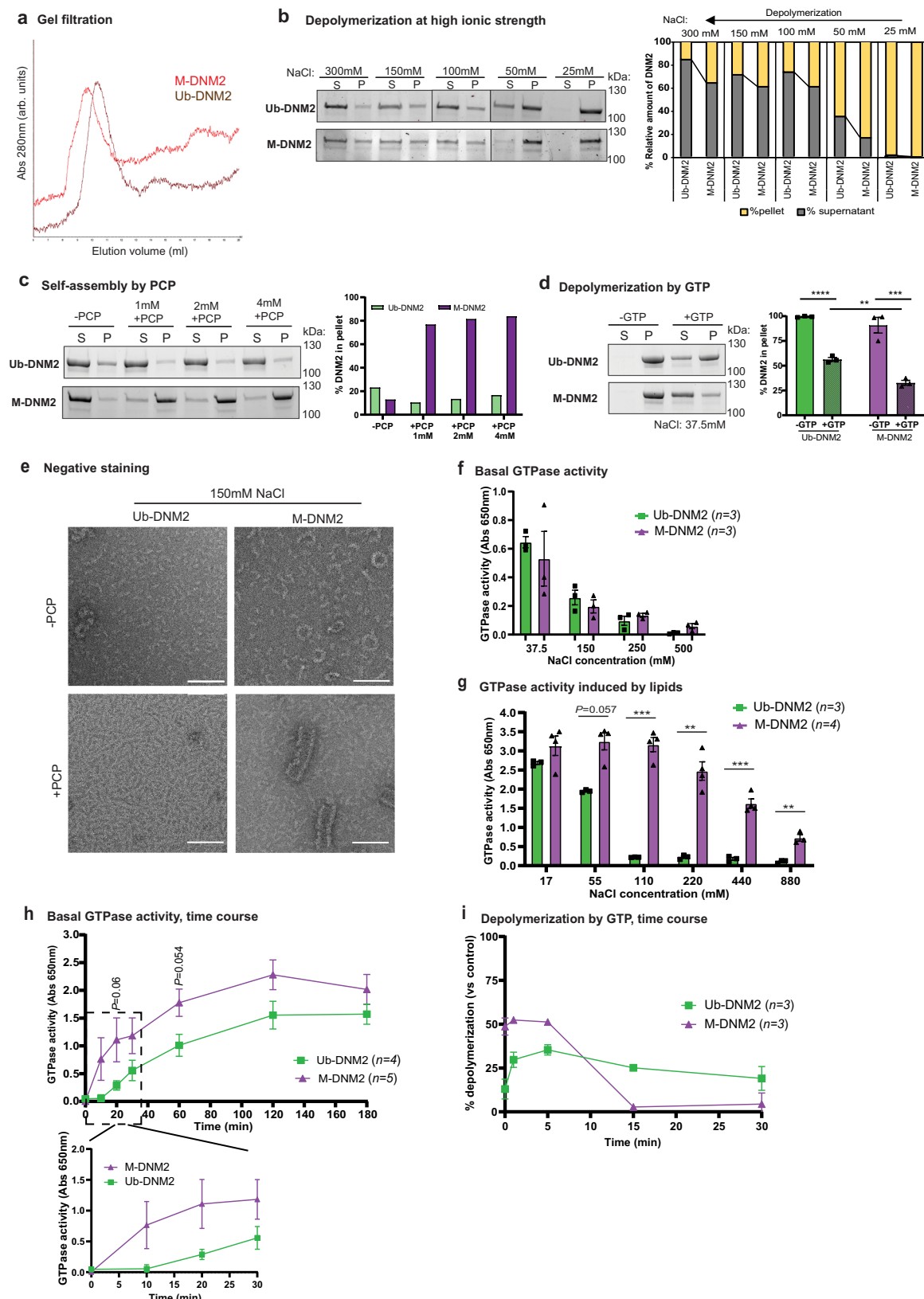

overall level. Previously, we showed decreasing the overall level of pan-DNM2 in *Mtm1*[−/y] mice rescued the disease[37,38] (Fig. 9). Taken together, our study provides new insights about CNM therapy targeting DNM2 and suggest that, contrary to first expectation, the importance of targeting and reducing the level of pan-DNM2 isoforms and not the muscle-specific M-DNM2 isoform for an efficient therapy. Thus,

increasing the muscle specificity of the treatment should not aim to target the muscle-specific DNM2 isoform but rather to engineer AAV serotypes or oligonucleotides with an increased tropism for muscle.

In conclusion our work present new insight on the molecular and cellular role of DNM2 isoforms in the muscle and provide new insights into the mechanistic and treatment direction in CNM.

**Fig. 7 | M-DNM2 forms larger oligomer structures, and it is functionally different to Ub-DNM2. a** Image of the elution of M-DNM2 and Ub-DNM2 following gel filtration chromatography. Column S200 10/300 was used, and salt concentration was 1.5 M NaCl. Absorbance (Abs) at 280 nm is represented as arbitrary units (arb. units). **b** Stain-free gel images for M-DNM2 and Ub-DNM2 fractioned in either supernatant (S) or in the pellet (P) after incubation at several salt concentrations. Right, proportion of DNM2 in S and P was calculated. **c** Stain-free gel images for M-DNM2 and Ub-DNM2 fractioned in either supernatant or in the pellet after incubation with GMP-PCP (PCP) or (**d**) after incubation at low salt concentration followed by depolymerization with GTP (2 mM GTP + 4 mM MgCl$_2$) (*n* = 3 independent experiments, **P = 0.0091, ***P = 0.0002, ****P < 0.0001 by 2-way ANOVA with Sidak's post hoc test). **e** Electron microscopy representative image of Ub-DNM2 and M-DNM2 at physiological salt concentration with or without PCP incubation. Scale bar = 100 nm. It was repeated in 3 independent experiments (with different protein productions). **f** GTPase activity measured in a phosphate release assay using malachite green dye with a reaction time of 90 min at 37 °C, expressed as absorbance at 650 nm. Measure for both DNM2 isoforms at different salt concentrations without lipids or (**g**) with 2-Diacyl-sn-glycero-3-phospho-L-serine (PS) lipid (110 mM (***P = 0.005), 220 mM (***P = 0.0023), 440 mM (***P = 0.0006), 880 mM (***P = 0.0024) by two-tailed unpaired *t* test at each salt condition). **h** Time course of basal GTPase activity at 37.5 mM NaCl during a maximum time of 180 min. Zoom of the first 30 min is shown below (two-tailed unpaired *t* test at each time point). **i** % of DNM2 depolymerization at 37.5 mM NaCl followed by addition of GTP (0.5 mM GTP + 2 mM MgCl$_2$) and incubation at different time points. The difference of DNM2 in the pellet between control condition (-GTP) and condition incubated with GTP is represented. **b, c** Result reproduced in independent experiments shown in Supplementary Fig. 13c and 14a, respectively. **d, f, i** 'n' in the figure indicates the number of independent experiments using protein from 2 different productions. Data are represented as mean values ± SEM. Source data are provided as a Source Data file.

**a** Tubulation assay: WT-DNM2 isoforms

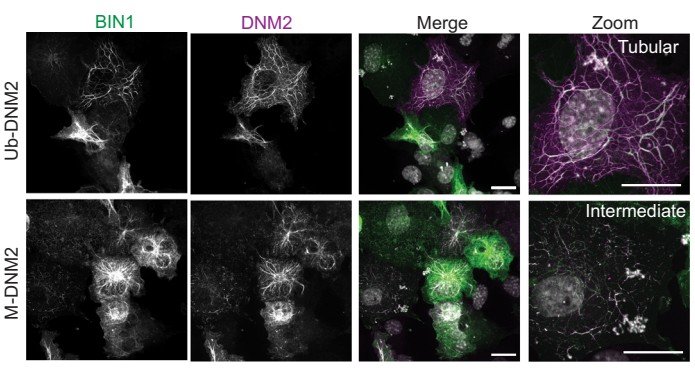
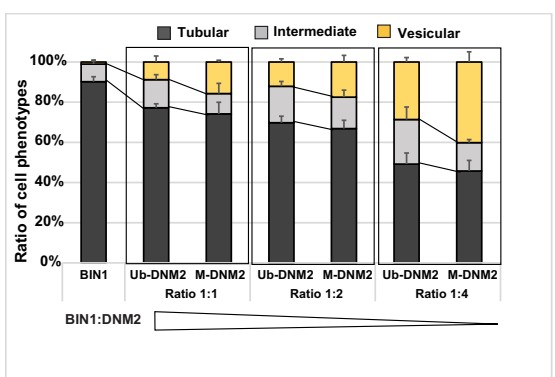

**b** Tubulation assay: S619L-DNM2 isoforms

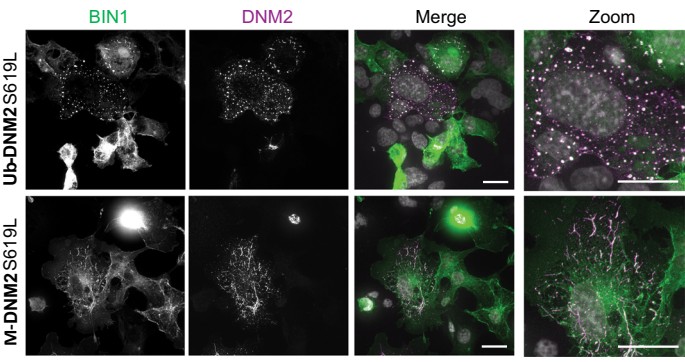
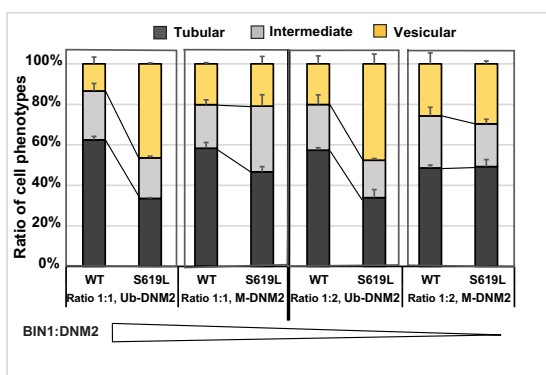

**Fig. 8 | Differential membrane fission activity of DNM2 isoforms. a** Fission activity in cellulo where BIN1-eGFP (green) and WT-DNM2 (magenta), M-DNM2 or Ub-DNM2, were co-expressed in COS-1 cells at incremental DNM2 concentration. Scale bar = 20 μm. **b** Fission activity in cellulo done as (**a**) co-expressing BIN1-eGFP (green) and the CNM mutant S619L-DNM2 (magenta). Scale bar = 20 μm. In both figures (**a, b**), percentage of each cell phenotype (tubular, intermediate or vesicular pattern) is shown at the right (average of 3 independent experiments). A representative image of the experiment at equal ratio of DNM2 and BIN1 is shown at the left. Data are represented as mean values ± SEM of a minimum of 25 cells/condition examined over 3 independent experiments. Source data are provided as a Source Data file.

## Methods

### Ethical statement

This research complies with all relevant ethical regulations. Animal experimentation was performed in agreement with the Animal Protection Law of France and was approved by the institutional ethical committee Com'Eth IGBMC-ICS (Illkirch, France) under the APAFIS project numbers 2019021109517421, 2019061316274324, 2016061019332648.

### Construct and reagents

Constructs. pAAV-DNM2 plasmids were generated as following described, cDNA of wild-type human DNM2 was cloned in pENTR1A vector (Invitrogen) and then recombined into pAAV-MCS vector (Agilent), using Gateway system, for expression control by CMV promotor[27,55]. pTL1 myc-His plasmids for human DNM2 mammalian expression and pEGFP-BIN1 plasmid (full length of human BIN1 isoform 8) have been previously used[21,54]. pGEX6P1 plasmid was used to produce SH3 of BIN1 in bacteria[54] and pVL1392 plasmids were used for human DNM2 recombinant protein production in baculovirus system as described in recombinant protein production section below[17]. Exon 12b inclusion, was generated by primer-direct PCR mutagenesis from WT construct using primers (forward, 5'CTGTTACTATACTGAGCAGCTGGTGACCTGTGCCCAGCAGAGGAGCACGC3'; reverse,5'CAGGT

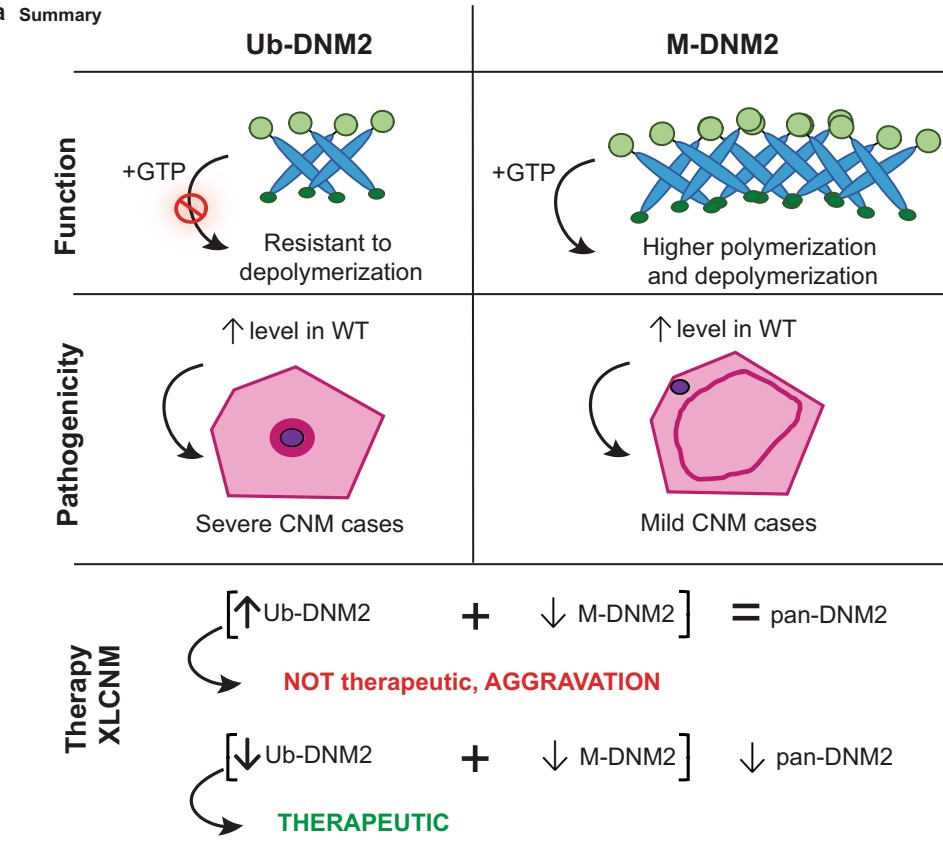

**a  Summary**

| Ub-DNM2 | M-DNM2 |

**Function**

+GTP → Resistant to depolymerization

+GTP → Higher polymerization and depolymerization

**Pathogenicity**

↑ level in WT → Severe CNM cases

↑ level in WT → Mild CNM cases

**Therapy XLCNM**

[↑Ub-DNM2  +  ↓ M-DNM2]  =  pan-DNM2
→ **NOT therapeutic, AGGRAVATION**

[↓Ub-DNM2  +  ↓ M-DNM2]  ↓ pan-DNM2
→ **THERAPEUTIC**

**Fig. 9 | Summary of the differential function and implications of DNM2 isoforms at pathophysiological and therapeutic levels.** Function: M-DNM2 was shown to form higher-order oligomer structures, while having faster GTP kinetics and higher sensitivity to GTP-induced depolymerization compared to Ub-DNM2 that was more resistant to depolymerization in presence of GTP. Pathogenicity: the differential molecular regulation of Ub-DNM2 and M-DNM2 underly that Ub-DNM2 exogenously expressed in mouse WT muscles triggered a severe CNM histology with centralized organelles, while expression of M-DNM2 led to a mild CNM phenotypes. Therapy: both isoforms have not an equivalent role in X-linked CNM (XLCNM) context as a splice switching strategy decreasing M-DNM2 while increasing Ub-DNM2 resulted in aggravation of the XLCNM phenotypes in mice, while maintaining an overall normal level of DNM2. Conversely, it was shown that decreasing the total level of pan-DNM2 resulted in the rescue of XLCNM mouse phenotypes[38].

CACCAGCTGCTCAGTATAGTAACAGTTGGCAAACCCGATGAAGTC3′). WT Ub-DNM2 cDNA used in this study did not contain exon 12b neither exon 13b (NCBI Reference Sequence: NM_004945). M-DNM2 and Ub-DNM2 plasmid contained same variant of exon 10. All constructs were verified by sequencing.

Antibodies. Primary antibodies used were dysferlin (ab15108, Abcam), α-actinin (A7811, Sigma-Aldrich), CHC (ab21679, Abcam), BIN1 (clone C99D, B9428, Sigma–Aldrich), TOMM20 (ab78547, Abcam). Mouse monoclonal Myc antibody (clone GE10) was produced in monoclonal antibody platform at IGBMC. Rabbit anti pan-DNM2 antibodies (R2680, R2865)[27] and pan-BIN1 (R3623, homemade anti-BIN1 SH3 domain, IGBMC)[35] were made onsite at the polyclonal facility at IGBMC. Novel M-DNM2 antibody (R3665-66) against 12b was also generated at the polyclonal facility at IGBMC. It is a rabbit antibody produced by immunization with a peptide including 12b amino acids. Serum was purified against peptide 12b and specificity determined by ELISA and by western blot against mouse and human M-DNM2 (Supplementary Fig. 1b). Alexa conjugated secondary antibodies were purchased from Invitrogen: Alexa Fluor® 488 Donkey Anti-Mouse IgG (H + L) (A21202), AlexaFluor® 647 Goat anti Rabbit IgG (H + L) antibody (A-21245) and AlexaFluor® 594 Goat anti mouse IgG (A-11005). Secondary antibodies against mouse and rabbit IgG, conjugated with horseradish peroxidase (HRP) were purchased from Jackson ImmunoResearch Laboratories: Peroxidase AffiniPure F(ab′)₂ Fragment Goat Anti-Rabbit IgG (H + L) (111-036-045), Peroxidase-conjugated affiniPure Goat anti mouse IgG+IgM (H + L) (115-036-068). Hoechst nuclear stain

is from Sigma–Aldrich. The specific dilutions of those primary and secondary antibodies are indicated in the respective method sections below as it depends on the application (Cell assay, Immunostaining of muscle sections and western blots). Further details about the validation of those antibodies can be found in the Reporting Summary.

Chemicals. The following products were purchased: ECL chemiluminescent reaction kit (Pierce), Lipofectamine3000 (Invitrogen), GMP-PCP (Jena Bioscience, NU-402-25), GTP (Sigma Aldrich), PS (Sigma), Sepharose (GE Healthcare), PreScission (GE Healthcare), FluorSaveTM reagent (Calbiochem), QuantiTect SYBR Green mix (Quiagen), DC Protein Assay (Bio-Rad), Ponceau S staining (Sigma–Aldrich).

**Animal models**
Animal experimentation was approved by the institutional ethical committee Com'Eth IGBMC-ICS (APAFIS project numbers 2019021109517421, 2019061316274324, 2016061019332648).

*Mtm1*[-/y] mouse line (129SvPAS) mice were previously generated and characterized[38,41]. *Dnm2*ex12b[−/−] mouse line (C57BL/B6NCrl) was generated as follows: the targeting vector was created with LoxP sites flanking exon 12b of *Dnm2*, then linearized, and electroporated into embryonic stem (ES) cells, derived from a C57BL/6 N, that were submitted to neomycin selection (G418). Identification of candidate recombinant clones were confirmed by long-range PCR and validated by southern blot. The selected ES clone was karyotyped and microinjected into BALB/C blastocysts that were implanted in pseudo

pregnant females. Resulting chimeric males were mated with wild-type C57BL/6NCrl females and germline transmission determined. Recombination was triggered constitutively using Rosa26-Cre C57BL/6N deleter line[56], kind gift from the Institute Clinic de la Souris (ICS). Genotyping to determined excision of exon 12b was performed with the following primers: forward, 5'GCTTTTAGAGCTCGGGTTTTAGA-GACG3' and reverse, 5'GGCTGGTGTCCTGCCTATTGATCTC3').

*Mtm1*[−/y] *Dnm2*ex12b[+/−] mice were obtained by breeding of female heterozygous *Mtm1*[+/−] mice with male heterozygous *Dnm2*ex12b[+/−] mice, and they were compared with *Mtm1*[−/y], *Dnm2*ex12b[+/−] and *Mtm1*[+/y] *Dnm2*ex12b[+/+] (WT) male littermates. In order to produce *Mtm1*[−/y] *Dnm2*ex12b[−/−], females *Mtm1*[+/−] *Dnm2*ex12b[+/−] from the previous crossing were bred with male heterozygous *Dnm2*ex12b[+/−] mice. This cross should yield 6 possible genotypes: *Mtm1*[−/y], *Mtm1*[−/y] *Dnm2*ex12b[+/−], *Mtm1*[−/y] *Dnm2*ex12b[−/−] (not obtained), *Mtm1*[+/y] *Dnm2*ex12b[+/+] (WT), *Mtm1*[+/y] *Dnm2*ex12b[+/−] (*Dnm2*ex12b[+/−]), *Mtm1*[+/y] *Dnm2*ex12b[−/−] (*Dnm2*ex12b[+/−]). Females *Mtm1*[−/+] *Dnm2*ex12b[−/−] were obtained in the offspring, and then phenotype in the study compared with female littermates.

Animals were housed in cages with access to food ad libitum and in temperature-controlled room (19 °C to 22 °C) ventilated by 100% fresh, non-recirculated air at the rate of 15 air changes per hour, humidity at 40–60% and 12:12-hour light/dark cycle. Animal housing room pressure is positive (0.5 to 2.55 Pa) to the corridor. Breeding animal were feed with diet SAFE®D03 (SAFE® diets), and transition after weaning to diet SAFE®D04 (SAFE® diets). *Mtm1*[−/y], *Dnm2*ex12b[+/−] and *Mtm1*[−/y] mice were feed with DietGel® 76A-72-07-5022 (w/ Animal Protein) (ClearH$_2$O), providing the gel food cups in the bottom of the cage, once they started to present ambulatory difficulties. Mice were euthanized by cervical dislocation and humanely sacrificed when required according to national and European legislations on animal experimentation.

### Recombinant protein production and gel filtration

Human SH3 domain of BIN1 with Glutathione S-transferase (GST) Tag (GST-SH3) was produced in *E. coli* BL21 using pGEX6P1 plasmid after induction with IPTG (1 mM) for 3 h at 37 °C. Then, cells were centrifuged at 7.500 g, lysate and GST-SH3 was purified using Glutathione Sepharose 4B beads (GSH-resin). The GST tag was cut by PreScission enzyme (GE Healthcare) on the beads and the flow through was recovered. Human M-DNM2 and Ub-DNM2 recombinant protein were produced in Sf9 insect cells-Novagen (Sigma–Aldrich Chimie S.a.r.l., Ref 71104-3) with the baculovirus system as follows[57]. Baculovirus were produced after transfection with DNM2 pVL1392 plasmids. Sf9 cells were infected with viruses and grown for 3 days at 27 °C and centrifuged at 2000 g for 10 min. DNM2 recombinant protein was purified with GST-SH3 of BIN1 bound to GSH-resin (GE Healthcare) as the SH3 domain captures full length dynamin 2 through a high affinity interaction with its PRD (column preparation described below)[58]. The analytical gel-filtration of DNM2 has been done using a column S200 10/300 and the buffer (20 mM Hepes, pH7.4; 5% Glycerol; 1.5 M NaCl; 1 mM EGTA; 1 mM DTT). Purity and quality of the proteins after elution was analyzed after separation with 12% SDS-PAGE gel follow by Coomassie staining.

### DNM2 binding to SH3 domain

For DNM2 pull down assay at increased SH3 domain of BIN1, different quantities of supernatant with recombinant GST-SH3 protein (0.16 ml, 0.4 ml and 1.6 ml) were applied in GSH-Sepharose column and incubated O/N and washed with buffer 300 mM NaCl, 50 mM Tris pH 8.0, 5 mM Imidazole, 1 mM DTT. Then, purified M-DNM2 and Ub-DNM2 (0.07 mg) recombinant protein was mixed with GST-SH3-sepharose resine, incubated 20 min and washed. The resine was mix with loading buffer and loaded in SDS-PAGE gel.

### Sedimentation assays

Sedimentation assays were conducted as follows[33,51].DNM2 (60 ng.µl[−1]) was incubated during 20 min at room temperature with different NaCl concentrations (300 mM, 150 mM, 100 mM and 50 mM) or different GMP-PCP concentrations (1 mM PCP + 2 mM MgCl$_2$, 2 mM + 4 mM MgCl$_2$ and 4 mM PCP + 4 mM MgCl$_2$). For GTP-induced depolymerization, DNM2 was firstly diluted in buffer with no salt (reaching 37.5 mM salt) and then GTP was added (0.5 mM GTP + 2 mM MgCl$_2$, 1 mM GTP + 2 mM MgCl$_2$ or 2 mM GTP + 4 mM MgCl$_2$) and incubated for 5 min at 37 °C (or 0,1, 5, 15 and 30 min for the time course). Assembled DNM2 was precipitated by centrifugation with 20,000 g for 30 min at room temperature. Resultant supernatant and pellet were denatured and separated in gradient precast Stain-Free gels (BioRad). Then, they were observed and scan in ChemiDoc MP Imaging System (BioRad) and images converted to Tiff using Image Lab software version 6.1.0. Relative level of DNM2 in supernatant and pellet was determined by densitometry using ImageJ software.

### Negative staining

5 µl of DNM2 (30 ng.µl[−1]) was deposited onto 300 meshs Cu/Rh grids covered with a carbon film (Euromedex CF300-CU-050) freshly plasma cleaned (Fischione 1070). After 60 s of absorption, each sample was stained with 2% uranyl acetate and observed by electron microscopy with a FEI Tecnai F20 microscope operating at a voltage of 200 kV equipped with a Gatan US1000 detector. Images were recorded using the SerialEM software version 3.8 at a nominal magnification of 50,000X, yielding a pixel size of 2.12 Å. For GMP-PCP (PCP) condition, same protocol was followed using higher concentration of protein (60–90 ng.µl[−1]) and incubating for 20 min at room temperature with 1 mM PCP and 2 mM MgCl$_2$. For each condition it was repeated at least with 2 different protein productions.

### GTPase activity

GTPase activity of DNM2 recombinant protein was measured with malachite green colorimetric assay[57]. 32 nM of DNM2 recombinant protein was incubated at different salt concentrations in the buffer 20 mM Hepes, pH7.4, 2.5 mM Mg2+), with or without 2-Diacyl-sn-gly-cero-3-phospho-L-serine (PS, 4 µg/ml) for 90 min at 37 °C. The concentration of GTP in the reaction mix was 0.5 mM in the basic assay without PS and 0.3 mM for the GTPase assay with PS. Reactions were terminated with Ethylenediaminetetraacetic acid (EDTA, pH7.4), and phosphate was detected by the addition of malachite green solution (2% w/v ammonium molybdate tetrahydrate, 0.15% w/v malachite green and 4 M HCl). For the time course, basic assay was done for a maximum time of 180 min and 37.5 mM NaCl. GTPase activity results were confirmed with at least 2 different protein productions. Absorbance at 650 nm was measured with a Synergy HTX Multimode Reader (Agilent BioTek) using software Gen5 version 2.07.

### Size exclusion chromatography and multi-angle light scattering (SEC-MALS)

Ub-DNM2 and M-DNM2 were passed through syringe filter Anotop 0.1 µm (Whatman) and 100ul of the samples at similar concentration were fractionated on the size exclusion column Superose 6 10/300 GL (GE Healthcare) connected to an Ettan microLC system (GE Healthcare), a MiniDAWN TREOS multi-angle light scattering detector (Wyatt Technology) and an Optilab T-rEX differential refractive interferometer (Wyatt Technology). The system was equilibrated with buffer in 20 mM HEPES (pH 7.4), 1.5 M NaCl, 1 mM EGTA, 1 mM DTT and 5% glycerol, and operated at 4 °C with a flow rate of 0.5 ml.min-1. Output data were analyzed using ASTRA 6.1 software (Wyatt Technology).

### Cell assays

COS-1 monkey fibroblast cell line was obtained from ATCC (Reference: CRL-1650) and authenticate by morphology. COS-1 cells were grown in Gibco Dulbecco's Modified Eagle Medium (DMEM) with 1 g/L glucose and supplemented with 5% Fetal Calf Serum (FCS) in a humified

incubator with 5% $CO_2$ at 37 °C. Expression plasmids were transfected using Lipofectamine 3000 according to manufacturer's instruction. Graphics are showing the results of minimum 3 independent experiments.

*In cellulo* tubulation assay. COS-1 cells were co-transfected with 0.5 μg BIN1-GFP plasmid and 0.5, 1 or 2 μg of M-DNM2-Myc, Ub-DNM2-Myc, M-DNM2 S619L-Myc or Ub-DNM2 S619L-Myc plasmids as described above. After 24 h of transfection, cells were washed with cold phosphate-buffered saline (PBS) and fixed with 4% paraformaldehyde (PFA) for 30 min. Then, permeabilization of the cells was done with 0.2% of Triton X-100 and unspecific binding sites were blocked with 5% bovin serum albumin (BSA) for 1 hour. Incubation with primary Myc antibody (diluted 1/500) was done O/N. The secondary antibody anti-mouse Alexa 594 (diluted 1/500) was incubated for 2 h. Nuclei was stained with Hoechst (1/400 dilution) incubated for 5 min and samples were mounted in FluorSaveTM reagent. Cells were observed on confocal microscope. For quantification of tubulation pattern, only cells co-transfected by both plasmids were considered and a minimum of 25 cells were analyzed per condition. We divided the cells in 3 phenotypes: tubular, intermediate and vesicular. Cells with punctuate pattern or "short" tubules with length below four times the tubule width are categorized as "vesicular", while cells exhibiting tubules that extended uninterrupted from nuclei to plasma membrane are considered as "tubular". Cells classified as "intermediate" displayed tubules longer than 4 times the width that are interrupted, so did not fully extend to the plasma membrane.

Immunofluorescence with vesicular enrichment. COS-1 cells were transfected with 1 μg of M-DNM2-Myc and Ub-DNM2-Myc. After 24 h, cells were treated with 0.5% Triton X-100, 2% PFA for 2 min at 37 °C to remove cytosolic DNM2 (vesicular enrichment fixation) as described before[59] followed by fixation in 4% PFA. Myc was used as primary antibody to detect transfected DNM2 together with Clathrin Heavy Chain (CHC) (1/300 dilution). Anti-mouse Alexa 594 and anti-rabbit Alexa 488 were used as secondary antibodies. Cells were observed on SP8X confocal microscope. Quantification of DNM2 vesicle area was done with Analyze Particle tool of Fiji setting a threshold from 0.15 μm². Then, it was calculated the average of the particle size for each cell.

## C2C12 cell culture differentiation

C2C12 mouse myoblast (ECACC, Ref: 91031101), authenticate by morphology, were subculture at 50% confluence and maintain in Dulbecco's Modified Eagle Medium (DMEM) (1 g/L glucose) supplemented with 20% Fetal Calf Serum (FCS) and Gentamicin (40 μg/ml). To differentiate the cells, 80% confluence should be reached, and medium changed to DMEM (1 g/L glucose) with 2% Horse Serum (HS) and Gentamicin (40 μg/ml). Once differentiation is initiated, half of cultured media was changed each two days and cells differentiated until 7 days.

## AAV transduction of tibialis muscle

AAV production and purification. Recombinant adeno-associated virus AAV2/1 were generated as described before[27,55]. In brief, AAV were generated by a triple transfection of HEK293T-derived cell line with the expression plasmid pAAV2-DNM2 (described pAAV-MCS plasmid in construct section), pXR1 containing rep and cap genes of AAV serotype 1 (UNC Vector Core), and pHelper encoding the adenovirus helper functions (Agilent). AAV vectors were harvested from cell lysate and purified by Iodixanol gradient ultracentrifugation (Optiprep™) followed by dialysis and concentration against Dulbecco's Phosphate Buffered Saline (DPBS) using centrifugal filters (Amicon Ultra-15 Centrifugal Filter Devices 100 K, Millipore). Viral titers were quantified by qPCR using the LightCycler480 SYBR Green I Master (Roche) and primers targeting CMV enhancer sequence. Titers are expressed as viral genomes per milliliter (vg/ml).

AAV intramuscular injection. AAV2/1 were injected in the Tibialis Anterior (TA) of 3-week-old male WT and Mtm1$^{-/y}$ 129/SvPAS mice as described before[55]. Mice were anesthetized by intraperitoneal injection of a solution of ketamine 20 mg/ml and xylazine 0.4% at 5 μl/g of body weight. Then, 15 μl of AAV encoding DNM2 (muscle and ubiquitous isoform) or empty AAV vectors were injected in the TA. Mice were sacrificed 2- or 4-weeks post injection. Viral quantity injected was optimized to get same levels of DNM2 overexpression with both isoforms, $6 \times 10^{10}$ to $3 \times 10^{11}$ vg/ml.

## Mouse phenotyping

The phenotyping tests listed below were conducted blinded.

Hanging test. Hanging ability was measured suspending the mice on a cage lid for a maximum time of 60 s. The time when the mouse fell off the cage was recorded for each trial. Three trials per mouse were performed, with 5 min of rest between trials. It is represented the average of the three trials for each mouse.

Disease Scoring System (DSS). It was designed to monitor the clinical appearance of the mice by following the evolution of six centronuclear myopathy features: body weight, hanging test ability, kyphosis, hindlimb position whilst walking, breathing ability, and ptosis. This scoring system was established in a previous study[38].

Rotarod. Coordination was measured placing the mice on a rotating rod (Bioseb), which accelerated from 4 to 40 rpm for 5 min. Three trials per mouse were performed with a resting time of 5 min. The latency to fall was measured.

Grip (4 paws). Grip strength test was performed placing the 4 paws on the grid of a dynamometer (Bioseb) and pulling by the tail in the opposite direction. The maximal strength for three trials was recorded for each mouse.

The mean of the 3 trials for each mouse was calculated for the tests listed above.

Actimetry. Spontaneous locomotor activity was measured in actimetric cages (Immetronic, Pessac, France) for 24 h with water and food pellet ad libitum. Specifically, each mouse was placed in an individual cage equipped with infra-red photo beam cells on the side walls of the cage to measure the animal movement horizontally and vertically.

Treadmill. Exercise/exhaustion test was used to determine maximal exercise capacity and endurance. Mice were placed on a treadmill line (Bioseb) with an inclination of 5 degrees angle and a speed of 25 cm/s. Both parameters increase with time up to a maximum speed of 41 cm/s, inclination of 15 degrees and time of 150 min. A grid that delivers a mild electric shock (less than 0.2 mA) is used to motivate the animal to run. Test is stopped when mice received more than 100 shocks/5 min or they stay 20 consecutives seconds next to the electrodes. It was recorded the maximal distance and latency performed for each mouse. Habituation was done the day before the test during 10 min, speed of 25 cm/s and 5 degrees angle.

Plethysmography. Spontaneous breathing pattern was measured in nonrestrained unstimulated mice using a whole-body barometric plethysmograph (EMKA Technologies) for 2 h. Breathing frequency is represented.

## In situ muscle force measurement

Tibialis Anterior (TA) muscle contraction was measured in response to nerve stimulation using the in situ whole animal system 1300 A (Aurora Scientific) and software 615 A: Dynamic muscle control and analysis software. Mice were anesthetized with intraperitoneal injections of domitor/fentanyl mix (2/0.28 mg/Kg), diazepam (8 mg/Kg) and fentanyl (0.28 mg/Kg). TA was partially excised and suture thread was used to attach the tendon to the isometric transducer. The sciatic nerve was stimulated by pulses of 2–200 Hz spaced by 30 s to measure maximal force. Specific maximal force was determined by dividing the maximal force with muscle cross sectional area calculated as wet muscle (mg) / optimal muscle length (mm) X mammalian muscle density (1.06 mg/mm³).

**Table 1 | List of primers used for RT-qPCR**

| Gene | Forward primer | Reverse primer |
|---|---|---|
| *Rpl27* | AAGCCGTCATCGTGAAGAACA | CTTGATCTTGGATCGCTTGGC |
| Pan-*Dnm2* | ACCCCACACTTGCAGAAAAC | CGCTTCTCAAAGTCCACTCC |
| M-*Dnm2* (+12b) | ACCTACATCAGGGAGCGAGA | TGTGACCAGCTCCTCAGTATAGA |
| Ub-*Dnm2* (−12b) | ACCTACATCAGGGAGCGAGA | GCTCCTCTGCTGGGCATT |
| *Dnm1* | AGATGGAGCGAATTGTGACC | GAATGACCTGGTTCCCTGAA |
| *Dnm3* | ATGCTCCGAATGTACCAAGC | GAGGGGAGCACTTATCGTCA |
| *Des* | CGACACCAGATCCAGTCCTAC | GCAATGTTGTCCTGATAGCCAT |

## Histology

Diaphragm was fixed in formaldehyde and embedded in paraffin, and 8 μm transversal and longitudinal sections were stained with hematoxylin and eosin (H&E).

TA muscles were frozen in liquid nitrogen-cooled isopentane and stored at −80 °C. 8 μm transversal sections were stained with H&E, succinate dehydrogenase (SDH), reduced nicotinamide adenine dinucleotide (NADH) and Periodic Acid-Schiff (PAS) to assess muscle fiber morphology, nuclear position, and distribution of mitochondria, reticulum, and glycogen. The images were recorded using the Nanozoomer 2HT slide scanner (Hamamatsu). Myofiber segmentation was performed using Cellpose 1.0 algorithm[60] and LabelsToROIS Fiji plugin[61] minimum Feret diameter (MinFeret) calculated using ImageJ. Fibers with mislocalized nuclei (centralized or internalized) and fibers with abnormal SDH distribution were counted using Cell Counter ImageJ plugin. Minimum of 800 fibers were measured for each muscle section and a percentage was calculated for each mouse.

## Immunostaining of muscle sections

For longitudinal immunostaining, TA muscles were fixed in PFA 4% for 24 h, then transferred to 30% sucrose and stored at −20 °C. For transversal staining, isopentane-frozen muscle was used, and muscle sections were fixed with PFA% prior immunostaining. 8 μm sections were permeabilized with 0.5% PBS-Triton X-100 and blocked with % bovine serum albumin (BSA) in PBS. The primary antibodies used, listed in material section, were diluted in 1% BSA as follows: DNM2 (R2680, 1/50), BIN1 (C99D, 1/50), CHC (1/100), Dysferlin (1/50), α-actinin (1/250), TOMM20 (1/200). The secondary antibodies were anti-mouse or -rabbit Alexa Fluor 488, 594 and 647 diluted 1/250 in 1% BSA. To detect F-actin, we used phallodin 555 probe diluted 1/1000 (Tebu Bio). Images were taken in Leica SP8-X confocal microscope and Z-projections are shown.

## Electron microscopy for morphology analysis

To perform morphological studies, pieces of TA muscle were fixed using 2.5% formaldehyde and 2.5% glutaraldehyde (Electron Microscopy Sciences) in 0.1 M cacodylate buffer (pH = 7.2; Sigma–Aldrich). Muscles were rinsed and followed by 1 h post-fixation in 1% osmium tetroxide [$OsO_4$] in H2O at 4 °C. Then samples were washed one time in distilled water. Samples were dehydrated with increasing concentrations of ethanol (10 min − 50%, 70%, 90% 95% and 3 × 100%), and embedded with a graded series of epoxy resin (ethanol/EPON 1/2, 2/3, 1/1). Samples were finally polymerized at 60 °C for 48 h. Ultrathin serials sections (70 nm) were picked up on grids, contrasted with uranyl acetate and lead citrate and examined using a Philips CM12 TEM electron microscope (Philips, FEI Electron Optics, Eindhoven, Netherlands) operated at 80 kV and equipped with an Orius 1000 CDD camera (Gatan, Pleasanton, USA).

## Electron microscopy for T-tubule staining

To perform T-tubules staining, pieces of TA muscle were fixed in 2.5% glutaraldehyde, 2.5% formaldehyde and 100 mM $CaCl_2$ in 0.1 M cacodylate buffer overnight at 4 °C. After cacodylate and water washing, samples were post-fixed for 1 h in 1% osmium tetroxide [$OsO_4$] reduced by 0.8% potassium hexacyanoferrate (III) [$K_3Fe(CN)_6$] in $H_2O$ at 4 °C. After extensive rinses in distilled water, muscles were then post-stained by 2% uranyl acetate for 2 h at room temperature. Then, samples were dehydrated with increasing concentrations of ethanol, and embedded with a graded series of epoxy resin (as detailed for morphology EM). Samples were finally polymerized and ultrathin serials sections (70 nm) were picked up on grids, contrasted with uranyl acetate and lead citrate. The grids were examined as described above for morphology analysis. Cell Counter ImageJ plugin was used to quantify % of misorientated T-tubules with longitudinal orientation (L-tubule), align to myofibril direction. At least 5 different fibers from 2–3 different mice were used for the quantification.

## RT-qPCR

RNA from the different tissues was extracted using TRI Reagent (Molecular Research Center) and reverse transcription was performed using SuperScript™ IV Transcriptase (ThermoFisher Scientific) with 0.5–1 μg of RNA. For quantitative PCR (qPCR), the cDNA was amplified performed in Lightcycler 480 (Roche) with software Version 1.2.9.11 using QuantiTect SYBR Green (Quiagen) and 0.1 μM of the following forward and reverse primers (Table 1). Primer specificity was determined through melting curve and Sanger-sequencing of PCR products. As reference gene, *Rpl27* was selected due to its stability in the different tissues and genotypes used in the study, as also described in previous publications[62,63].

## Protein extraction and Western blot

Tissues were lysed in Radioimmunoprecipitation assay (RIPA) buffer supplemented with 1 mM PMSF, 1 mM DTT and complete mini EDTA-free protease inhibitor cocktail (Roche) using a Precellys tissue homogenizer (Bertin Technologies). DC Protein Assay was used to calculate protein concentration. Protein (10 μg) was denatured in 5× Lane Marker Reducing Buffer (Thermo Fisher Scientific) and then separated in 7, 8 or 10% SDS-PAGE gel. The gel was transferred on nitrocellulose membrane using Transblot Turbo RTA transfer kit (Bio-Rad). Protein loading was determined by Ponceau S staining[64]. Membranes were blocked for 1 h in Tris-Buffered Saline (TBS) buffer containing 5% nonfat dry milk and 0.1% Tween-20. Membranes were incubated with primary antibodies 1 h room-temperature in 5% milk and subsequently, with secondary antibodies coupled to horseradish peroxidase during 1 h at room-temperature. Primary antibodies used were pan-DNM2 (R2865 homemade anti-PRD, 1/700), M-DNM2 (homemade anti-12b, 1/200), pan-BIN1 (R3623, homemade anti-BIN1 SH3 domain, 1/1000) and CHC (1/1000). Secondary antibodies were anti-rabbit HRP or anti-mouse HRP (1/5000). Images were recorded with Amersham Imager 600 (GE Healthcare Life Sciences) and band intensity was determined by densitometry using ImageJ and divided by intensity of protein loading. Uncropped versions of the blots are shown in Supplementary Fig. 16.

## Cloning

To clone *DNM2* transcripts expressed in human skeletal muscle tissue, RT-PCR was performed with RNA from human skeletal muscle purchased from Clontech (NC9946721) and the PCR product was purified with NucleoSpin Gel and PCR clean-up kit (Macherey-Nagel). The purified PCR product was cloned with pGEM®-T Easy Vector system (Promega) and ligation was done according to manufacturer's instruction. Briefly, 3 µl of PCR product were incubated at 4 °C O/N with ligation buffer including T4 DNA ligase and pGEM®-T Easy Vector. Next, 5 µl of ligation product was transformed in DH5 α competent cells following a heat shock protocol and then added to LB plates with ampicillin, IPTG (isopropyl-β-d-thiogalactopyranoside) and X-Gal. The day after, white colonies were picked and grown in LB with ampicillin O/N, follow by purification of the plasmid with NucleoSpin Plasmid kit (Macherey-Nagel). Finally, the purified plasmid was sent to sequence with primer pairs used for RT-PCR. 32 clones were sequenced.

## RNA sequencing

RNA extraction and RT using TA muscle from WT (129SvPas) mice at different ages was done as described above. To perform RNA sequencing, library preparation was done with TruSeq Stranded mRNA Sample Preparation Kit (Illumina, San Diego, USA) and polyA selection, sequenced as paired-end 100 bp reads on a Hiseq4000.

## Transcriptomic analysis

Reads were preprocessed using cutadapt (version 1.10) to remove adapter, polyA and low-quality sequences (Phred quality score below 20). Reads shorter than 40 bases were excluded from further analysis. Reads were mapped onto the GRCm39/mm39 version assembly of *Mus musculus* genome using STAR version 2.5.3. Transcripts were quantified by the Cuffllinks version 2.2.1[65] with default parameters. Differential gene expression analysis was performed using Cuffdiff version 2.2.1.

## Statistical analysis

All data were verified for normal distribution using Shapiro-Wilk test and for homoscedasticity with the Brown-Forsythe test. Significant changes for normal data were examined using two-tailed unpaired Student's *t*-test for 2 group comparison, or 1-way ANOVA followed by Tukey's post hoc test in case of multiple group analyses. Welch's correction was applied in case of difference between standard deviations. In case of not-normally distributed data, Mann–Whitney test was used to compare 2 groups or nonparametric 1-way ANOVA (Kruskal-Wallis) with Dunn's post hoc test was done for more than 2 groups comparison. A *P* value less than 0.05 was considered significant. All statistical tests used were 2-sided. Multiple time points were analyzed separately. The significant difference of birth ratio was assessed by chi-squared test. For survival curves, the log-rank (Mantel-Cox) test was performed. Charts show individual points with additional lines indicating mean ± SEM. Significant differences are illustrated as *$P < 0.05$, **$P < 0.01$, ***$P < 0.001$ and ****$P < 0.0001$ and exact *P* value indicated in the figure legend. Curves and graphs were made using GraphPad Prism or Microsoft Excel software.

## Reporting summary

Further information on research design is available in the Nature Portfolio Reporting Summary linked to this article.

## Data availability

The authors declare that all data supporting the findings of this study are available within the paper and its supplementary information files. Source data from Main and Supplementary figures generated in this study are provided with this paper as a Source Data file. Uncropped and unprocessed western blot and Stain-Free gels are provided in the Supplementary Fig. 16 in the "Supplementary Information File". A reporting summary for this article is available as a "Supplementary Information File".

RNA sequencing data supporting this study has been deposited into the Gene Expression Omnibus (GEO) database (NCBI). It is accessible under GEO Series accession number GSE214621. GRCm39/mm39 version assembly of Mus musculus genome is accessible under this link: https://www.ncbi.nlm.nih.gov/assembly/GCF_000001635.27. Source data are provided with this paper.

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

## Acknowledgements

We thank IGBMC platforms for animal housing, imaging, electron microscopy, integrated structural biology and histology. Chaouki Bam'Hamed, Nadine Banquart-Ott, Chadia Nahy, Marion Ciancia, Christine Kretz, David Reiss for animal care and phenotyping, Jean-Luc Weickert and Florence Granger for technical assistance, Pascal Kessler for imaging, Suzie Buono, Marion Depla, Catherine Koch, Valentina Lionello, Anne Robé, Catherine Birck and Roberto Silva-Rojas for scientific advice and suggestions. The creation of the *Dnm2*ex12b floxed mouse was done with Phenomin-ICS (Illkirch, France). This study was supported by INSERM, CNRS, Strasbourg University, ANR-10-LABX-0030-INRT, ANR-10-IDEX-0002-02, ANR Dynather (ANR-18-CE17-0006-02) received by J.L.; French Infrastructure for Integrated Structural Biology (FRISBI) ANR-10-INSB-05-01 and INSTRUCT-ERIC supported work by P.P.C.; and direct funding from Dynacure. RGO was supported by Cifre (ANRT, 2017/1270).

## Author contributions

B.S.C. and J.L. conceived and supervised the project. R.G.O., E.E., S.D., B.S.C., J.L. designed the experiments. R.G.O. performed most of the experiments with the help of E.E., S.D., M.G., X.M.M., C.C., C.S., N.D., P.P.C. M.O.A. and P.K. contributed with materials. R.G.O., E.E., M.G., B.S.C., J.L. analyzed the data. R.G.O., J.L. and B.S.C. wrote the manuscript with input of all coauthors.

## Competing interests

B.S.C. and J.L. are co-founders of Dynacure, R.G.-O., M.G. and B.S.C. are employees of Dynacure. The remaining authors declare no competing interests.
