## [Peer Review File · Nature Communications]

Differential impact of ubiquitous and muscle dynamin 2 isoforms in muscle physiology and centronuclear myopathyREVIEWER COMMENTS

Reviewer #1 (Remarks to the Author):

The mechano-chemical GTPase Dynamin 2 is a key player in membrane scission and other cellular membrane remodelling events in eukaryotes. Specific mutations in dynamin 2 are associated with a severe muscle disorder, Centronuclear Myopathy (CNM). Why these mutations affect mostly the muscle, but not other tissues, is not known and may be related to tissue-specific splice isoforms of dynamin. The current manuscript seeks to decipher this hypothesis by characterizing specific splice variants of dynamin 2 expressed either ubiquitously (Ub-Dyn2) or in a muscle-specific manner (M-Dyn2). Targeted deletion of exon 12 specific for M-Dyn2 did not lead to an obvious phenotype in mice. While overexpression of Ub-Dyn2 in muscle induced major disturbances in muscle organisation, overexpression of M-Dyn had less severe effects. In combination with MTM1 loss of function, the loss of exon 12 in dynamin and, consequently, the switch to the Ub-Dyn isoform induced a severe muscular phenotype. Biochemical analyses indicate an increased propensity of M-Dyn2 vs. Ub-Dyn2 to assemble into higher order oligomers.

This study addresses an interesting open question in the dynamin field, e.g. the role of tissue-specific dynamin 2 splice variants and their link to the development of tissue-specific disease. The manuscript is well drafted and contains a lot of experimental work, including the generation and thorough characterization of several mouse models, the development of a specific reagent to recognize the muscle-specific M-Dyn2 and a biochemical comparison of M-Dyn2 vs Ub-Dyn2. As far as I can judge, the experiments are carefully conducted, including sound statistical analyses, and the conclusions are well supported by the experiments, resulting in a coherent story. My main concern is the somewhat unclear connection of the conducted biochemical experiments to the observed disease phenotype.

Major

The biochemical characterization indicates a higher propensity of M-Dyn2 for oligomerization, increased GTPase rates and membrane scission compared with Ub-Dyn2, indicating a higher activity of M-Dyn2. In light of this, it is puzzling that over-expression of Ub-Dyn2 has apparently a more devastating effect on muscle structure and physiology than overexpression of M-Dyn2. Thus, the molecular difference of Ub-Dyn2 and M-Dyn2 underlying disease progression in muscle remains somehow vague. Some additional biochemical experiments may shed further light on this. For example,

- Fig. 7D: It is astonishing that M-DNM2 has an increased propensity to assemble in the presence of GMPPCP and, at the same time, an increased GTP-dependent disassembly rate. Disassembly may be related to the increased GTPase activity of M-Dyn2. To further characterize this idea, I would suggest repeating the sedimentation experiments in Fig. 7D with different time points and examine in parallel the GTP turnover in these reactions (for example, immediately after GTP addition, when 50% GTP is hydrolysed and when all GTP is turned over for both mutants). This analysis may reveal an increased turnover rate of the M-Dyn2 oligomers as potential cause for their reduced toxicity in muscle.

- Fig. 7H: It may be also envisaged that the disease mutations have a more severe impact on Ub-DNM2 than on M-DNM2. To test this idea, I would suggest including the S619L mutation in the context of M-DNM2 in this analysis.

Minor:

Fig. 7A: These data merely show that M-DNM2 has a larger hydrodynamic radius than Ub-DNM2, which may be due to higher order assembly of M-DNM2 or, as likely, due to the PH domains having more flexibility because of their extended linkers. With a gelfiltration coupled to a right angle laser scattering device, the authors could distinguish between these scenarios. Please mention the used gelfiltration column and salt concentration in the legend.

Fig. 7F: Please add some more specifics for the GTPase assay in the legend (end point measurement after how many minutes, phosphate release assays detected by malachite green, etc.)

Typos, etc

Line 67 helicoidal – helical ?

Line 70 as microtubule-associated proteins

Line 77 Differential splicing was found to impact Golgi targeting of DNM2

Line 178 caused

Figure 7B: strength

Figure 7C: depolymerization

line 1014, followed by ultracentrifugation and pellet (P) and supernatant (S) fractionation.

Reviewer #2 (Remarks to the Author):

The authors present exciting new insights on the role of the molecular and cellular role of dynamin 2 isoforms through experimentation in mouse and cell models. They show biochemical and functional differences of a muscle predominant isoform and that upregulation of the ubiquitous, not the muscle-specific isoform worsens myotubular myopathy, a fatal human muscle condition. This has important implications for the design of targeted therapies in myotubular myopathy. I have some comments for consideration:

1. The tissue expression of the muscle and the ubiquitous dynamin 2 isoforms is mainly based on RNA and protein data in mice. The authors mention briefly that similar isoforms have been found in humans according to the Gtex database. This might not be sufficient to make assumptions about the abundance of these isoforms in human muscle. I would strongly encourage the authors to specifically investigate the expression pattern in human muscle tissue.

2. In line 279/280 "severe neonatal CNM phenotype" following the intramuscular injection of AAV into 4-week old mice is an overinterpretation and should be tuned down.

3. In order to show the pathological consequence of the muscle and ubiquitous isoforms in muscle, the authors injected wildtype animals with the corresponding AAV constructs and showed that overexpression of the ubiquitous DNM2 isoform leads to a more profound pathology. It would be interesting to see whether similar experimentation in the XLCNM mouse (MTM1 deficient) would lead to an exacerbation of the phenotype

Reviewer #3 (Remarks to the Author):

This work is focused on the isoforms and functions of Dynamin 2 (DNM2) and their roles in centronuclear myopathies with a major focus on the ubiquitous (Ub-DNM2) and muscle-specific DNM2 isoforms (M-DNM2). The authors report that M-DNM2 forms large oligomers that sequester BIN1 whereas increasing Ub-DNM2 correlates with severe CNM forms and increasing M-DNM2 overexpression leads to hallmarks of mild CNM. Dnm2 splice switching from M-Dnm2 towards Ub-Dnm2 aggravates the phenotypes of severe X-linked CNM caused by loss-of-function of the MTM1. This work is focused on an uninteresting and clinically relevant topic, is well carried out and well described.

My main comment is that the work is rather amorphous and that it is often unclear what the major conclusions are. At least for this reviewer the work does not well 'gel'. Perhaps a summary figure capturing the major conclusions would be helpful.

The authors suggest that their findings support the importance of therapeutically targeting all DNMT2 isoforms but it is unclear how this could be done. Can you be more specific?

Response to Referees Letter

RE: Differential impact of ubiquitous and muscle dynamin 2 isoforms in muscle physiology and centronuclear myopathy (Manuscript reference: NCOMMS-21-35521)

We thank the reviewers for their constructive comments and suggestions. We have carefully addressed all concerns and provide point-by-point responses to these comments in the following sections. We have performed all experiments asked and in consequence, made major changes and additions to the revised manuscript, as detailed below, in main and supplementary figures, figure legends, discussion and methods. All modifications are highlighted in the revised manuscript using the track-changes option and main additions in the text have been also indicated in the following response. We believed these changes and additional work have significantly improved our manuscript and strengthened the connection between the biochemical and cellular experiments and the observed *in vivo* phenotypes. We have also clarified the significance of our work with changes in the text, especially in the discussion, and the addition of a summary figure to present our findings clearer to a non-specialized audience. We hope that the editor and reviewers will now find the revised manuscript suitable for publication.

REVIEWER COMMENTS

Reviewer #1 (Remarks to the Author):

The mechano-chemical GTPase Dynamin 2 is a key player in membrane scission and other cellular membrane remodelling events in eukaryotes. Specific mutations in dynamin 2 are associated with a severe muscle disorder, Centronuclear Myopathy (CNM). Why these mutations affect mostly the muscle, but not other tissues, is not known and may be related to tissue-specific splice isoforms of dynamin. The current manuscript seeks to decipher this hypothesis by characterizing specific splice variants of dynamin 2 expressed either ubiquitously (Ub-Dyn2) or in a muscle-specific manner (M-Dyn2). Targeted deletion of exon 12 specific for M-Dyn2 did not lead to an obvious phenotype in mice. While overexpression of Ub-Dyn2 in muscle induced major disturbances in muscle organisation, overexpression of M-Dyn had less severe effects. In combination with MTM1 loss of function, the loss of exon 12 in dynamin and, consequently, the switch to the Ub-Dyn isoform induced a severe muscular phenotype. Biochemical analyses indicate an increased propensity of M-Dyn2 vs. Ub-Dyn2 to assemble into higher order oligomers.

This study addresses an interesting open question in the dynamin field, e.g. the role of tissue-specific dynamin 2 splice variants and their link to the development of tissue-specific disease. The manuscript is well drafted and contains a lot of experimental work, including the generation and thorough characterization of several mouse models, the development of a specific reagent to recognize the muscle-specific M-Dyn2 and a biochemical comparison of M-Dyn2 vs Ub-Dyn2. As far as I can judge, the experiments are carefully conducted, including sound statistical analyses, and the conclusions are well supported by the experiments, resulting in a coherent story. My main concern is the somewhat unclear connection of the conducted biochemical experiments to the observed disease phenotype.

Major

The biochemical characterization indicates a higher propensity of M-Dyn2 for oligomerization, increased GTPase rates and membrane scission compared with Ub-Dyn2, indicating a higher activity of M-Dyn2. In light of this, it is puzzling that over-expression of Ub-Dyn2 has apparently a more devastating effect on muscle structure and physiology than overexpression of M-Dyn2. Thus, the molecular difference of Ub-Dyn2 and M-Dyn2 underlying disease progression in muscle remains somehow vague. Some additional biochemical experiments may shed further light on this. For example,

- Fig. 7D: It is astonishing that M-DNM2 has an increased propensity to assemble in the presence of GMPPCP and, at the same time, an increased GTP-dependent disassembly rate. Disassembly may be related to the increased GTPase activity of M-Dyn2. To further characterize this idea, I would suggest repeating the sedimentation experiments in Fig. 7D with different time points and examine in parallel the GTP turnover in these reactions (for example, immediately after GTP addition, when 50% GTP is hydrolysed and when all GTP is turned over for both mutants). This analysis may reveal an increased turnover rate of the M-Dyn2 oligomers as potential cause for their reduced toxicity in muscle.

Response: We appreciate the reviewer's comment and found this hypothesis coherent to the data already presented in the manuscript showing M-DNM2 increased GTP-dependent depolymerization rate and how this could be a potential cause to explain a lower toxicity to the muscle. To further explore this hypothesis and make clearer the connection between the *in vitro* and *in vivo* data, main concern of the reviewer, we have performed the experiment suggested: sedimentation experiment to assess DNM2 depolymerization after GTP addition in parallel with the analysis of GTP turnover rate at different time points under the same conditions (low salt: 37.5 mM and GTP concentration of 0.5 mM).

- When setting up the sedimentation assay with different time points, we observed a faster depolymerization in the case of M-DNM2 versus Ub-DNM2, when GTP was added to preassembled DNM2 (at low salt). Even at time 0 (when GTP is added to preassembled DNM2 on ice), partial reversal of M-DNM2 from the pellet to the supernatant fraction was observed, highlighting M-DNM2 was more sensitive to GTP-induced depolymerization than Ub-DNM2 at this timepoint. We observed with time, after 15 min, M-DNM2 was reverting to the pellet (polymerized state), and this could be explained by DNM2 oligomerization once most GTP is consumed. In line with the previous result showing M-DNM2 is more sensitive to GTP-induced depolymerization than Ub-DNM2 (Fig. 7d), the ratio of GTP-induced depolymerization of M-DNM2 oligomers was higher than for Ub-DNM2 oligomers (48% vs 35% at 5 min). Additionally, M-DNM2 achieved maximum GTP-induced depolymerization at time 1 min (T1; 52%) while maximum GTP-induced depolymerization for Ub-DNM2 was reached at time 5 min (T5; 35%). These new sedimentation assay results are shown now in **Figure 7i**, and how the percentage of depolymerization is calculated from analysis of sedimentation assay gels is shown in **Supplementary Figure 15b**.

- In parallel we have examined the GTP turnover with time by a phosphate release assay using malachite green dye (as used for Fig. 7f) with the same conditions as for the sedimentation assay. We observed faster kinetics and GTP turnover rate for M-DNM2 compared to Ub-DNM2. In detail, M-DNM2 hydrolyzed 50% of the GTP faster (~20 min) in comparison with Ub-DNM2 (~40 min). This result is now shown in **Figure 7h** and mentioned percentage can be observed in **Supplementary Figure 15a**.

In conclusion, we observed faster GTP hydrolysis kinetic of M-DNM2 correlating with faster GTP-dependent depolymerization rate of M-DNM2 oligomers, compared to Ub-DNM2. This is in agreement with the reviewer's hypothesis. Taken together with our other data, it shows that M-DNM2 has an increased oligomerization rate, GTPase activity, and GTP-induced depolymerization than Ub-DNM2. This could explain the reduced toxicity of this isoform in mouse muscles when compared to Ub-DNM2. Indeed, the literature reports several CNM mutants as R369W, R465W or S619L presenting resistance to GTP-induced depolymerization and how this could be linked to muscle atrophy (Wang *et al.* 2010, Chin *et al.* 2015, Fujise *et al.* 2021, Shan-Shan *et al.* 2020). The fact that M-DNM2 has an opposite sensitivity to GTP-induced depolymerization than the CNM mutants, compared to Ub-DNM2, is most probably at the basis of its reduced toxicity *in vivo*.

We have also mentioned these new results in the Result section, sentence in line 370-371: "In addition, M-DNM2 presented faster GTP hydrolysis as well as faster GTP-induced depolymerization compared with Ub-DNM2 (Fig. 7h-i, Supplementary Fig. 15 a-b)"

As these novel results are also important to highlight the molecular difference between the isoforms that may underlay the differential disease progression observed *in vivo*, we highlighted it in the discussion as well as in the novel summary figure (Figure 9).

- Fig. 7H: It may be also envisaged that the disease mutations have a more severe impact on Ub-DNM2 than on M-DNM2. To test this idea, I would suggest including the S619L mutation in the context of M-DNM2 in this analysis.

Response: We thank the reviewer for the suggestion to test if a CNM disease mutation has a different impact on membrane fission depending on the DNEM2 isoform.

As suggested by the reviewer, we have performed the *in cellulo* tubulation assay similarly to the experiment in Fig. 8a (previously Fig. 7h), expressing the disease mutation S619L in both M-DNM2 and Ub-DNM2 isoforms. Expression of BIN1 creates membrane tubules emanating from the plasma membrane (Spiegelhalter *et al.* 2010), and co-expression of DNEM2 constructs could fission these tubules depending on transfected amount.

First, we confirmed our previous data that WT M-DNM2 expression correlated with a higher proportion of cells with vesicles or short tubules compared to WT Ub-DNM2. The DNEM2-S619L mutant has been already tested in a Ub-DNM2 isoform in the previous version of our manuscript (previous Fig. 7h) and in the article Gibbs *et al.* 2014, and resulted in the disruption of BIN1-membrane tubules into vesicular patterns. Now we found the addition of the S619L mutation caused an increase in the membrane fission activity of Ub-DNM2 compared to M-DNM2.

Also, we observed a ~30% increase in cells with vesicular patterns with Ub-DNM2-S619L compared to Ub-DNM2-WT, while nearly no effect of the S619L mutation could be noted in the M-DNM2 isoform (~3%). These observations were seen at 2 different ratios of BIN1:DNM2 (increasing DNM2 concentration).

In conclusion, as suggested by reviewer 2, we have now shown that the disease mutation S619L exacerbates the membrane fission capacity of Ub-DNM2 compared to M-DNM2, sustaining the Ub-DNM2 isoform is the main contributors of CNM pathogenesis *in vivo* as it is more sensitive to the CNM mutation.

We moved our previous data on WT DNM2 isoforms to **Figure 8a** and included the novel results on the mutants as **Figure 8b** and in sentence number 381-385 in the text from result section:

“Of note, a CNM mutation (DNM2-S619L) in the Ub-DNM2 isoform strongly increased tubules fission by ~30% (Fig. 8b). We thus investigated if the addition of the S619L mutation had the same impact in a M-DNM2 isoform context and observed only a slight increase in the proportion of cells with vesicular pattern (~3%) compared to its equivalent WT form (Fig. 8b).”

The abstract (line 27-28) and discussion (line 456-458) was updated based on these novel findings:

“*In vitro*, the fission activity of Ub-DNM2 was more sensitive to a CNM mutant than M-DNM2.”

“Moreover, we found the addition of the CNM mutation S619L in Ub-DNM2 exacerbates its membrane fission properties in cells compared to the impact in M-DNM2, which may be already in a pre-activated conformation.”

Minor:

Fig. 7A: These data merely show that M-DNM2 has a larger hydrodynamic radius than Ub-DNM2, which may be due to higher order assembly of M-DNM2 or, as likely, due to the PH domains having more flexibility because of their extended linkers. With a gel filtration coupled to a right angle laser scattering device, the authors could distinguish between these scenarios. Please mention the used gel filtration column and salt concentration in the legend.

Response: In order to better elucidate if the larger hydrodynamic radius of M-DNM2 compared to Ub-DNM2, observed with gel filtration, is due to higher order assembly we performed Size Exclusion Chromatography couple with Multi-Angle Light Scattering (SEC-MALS). SEC-MALS has been already used in dynamin field to assess the impact of dynamin mutations on its assembly properties (Ramachandran, R. *et al.* 2017, Srinivasan, S. *et al.* 2016).

We observed differences between Ub-DNM2 and M-DNM2 after SEC-MALS analysis under similar conditions. The main peak for Ub-DNM2 has an elution volume of 12.5 ml and average molecular masses ranging from 200 to 350 kDa. The main peak for M-DNM2 is eluted around 11.5 ml with average molecular masses ranging from 400 to 470 kDa. This technique also confirmed the average hydrodynamic radius of M-DNM2 is larger (6.5 nm) than the one of Ub-DNM2 (3.5 nm).

We have included the results in **Supplementary Fig. 13b** showing a representative SEC profile and the SEC-MALS analysis of average molecular masses. A sentence with this main finding was added to number 337-343 in the text: “To assess if this shift in gel filtration corresponds to higher order assembly of M-DNM2, both DNM2 isoforms were analyzed by size-exclusion chromatography combined with multiangle light scattering (SEC-MALS). SEC elution profile showed again Ub-DNM2 is eluting later than M-DNM2 (12.5 ml vs. 11.5 ml) and MALS confirmed this is due to the presence of larger oligomers of M-DNM2 with average molecular mass from 400 to 470KDa while Ub-DNM2 formed oligomer species from 200 to 350KDa (Supplementary Fig. 13b).”

In conclusion, this methodology showed that the higher hydrodynamic radius observed by gel filtration is most probably due to higher order assembly of M-DNM2 compared to Ub-DNM2 under the same conditions. This is also in line with negative staining observations of both proteins under similar conditions (Supplementary Fig. 14c), where M-DNM2 formed higher-order assembly (horseshoes) compared to lower oligomeric assembly of Ub-

DNM2. We do not however exclude that the PH domain may have more flexibility because of the extended linker in M-DNM2 compared to Ub-DNM2. To investigate changes in domain flexibility, other approaches such as Hydrogen–Deuterium Exchange coupled with Mass Spectrometry (HDX-MS) have been used with DNM1 (Srinivasan, S. *et al.* 2016). We do not have access to this technology and DNM2 has different properties, thus this future investigation will require long steps of optimization in collaboration with experts in the field.

Finally, as also requested by the reviewer, the gel filtration column and salt concentration were indicated in figure 7a legend, line 1078: “Column S200 10/300 was used, and salt concentration was 1.5 M NaCl”.

Fig. 7F: Please add some more specifics for the GTPase assay in the legend (end point measurement after how many minutes, phosphate release assays detected by malachite green, etc.)

Response: More specifics for the GTPase assay were added to the figure 7f legend (line 1089-1094): “(f) GTPase activity was measured in a phosphate release assay using malachite green dye with a reaction time of 90 minutes at 37°C, and expressed as absorbance at 650 nm of both isoforms of DNM2 at different salt concentrations without lipids or (g) with 2-Diacyl-sn-glycero-3-phospho-L-serine (PS) lipid.”

More details can also be found in material and method section GTPase activity, line 603-612.

“GTPase activity of DNM2 recombinant protein was measured with malachite green colorimetric assay as previously described (Quan *et al.* 2005). 32 nM of DNM2 recombinant protein was incubated at different salt concentrations in the buffer 20 mM Hepes, pH7.4, 2.5 mM Mg²⁺, with or without 4 µg/ml of 2-Diacyl-sn-glycero-3-phospho-L-serine (PS) for 90 minutes at 37°C. The concentration of GTP in the reaction mix was 0.5 mM in basic assay without PS and 0.3 mM for GTPase assay with PS. Reactions were terminated with Ethylenediaminetetraacetic acid (EDTA, pH7.4), and phosphate was detected by the addition of malachite green solution (2% w/v ammonium molybdate tetrahydrate, 0.15% w/v malachite green and 4 M HCl). For the time course, basic assay was done for a maximum time of 180 minutes and 37.5 mM NaCl. GTPase activity results were confirmed with at least 2 different protein productions.”

Typos, etc

Line 67 helicoidal – helical ?

Line 70 as microtubule-associated proteins

Line 77 Differential splicing was found to impact Golgi targeting of DNM2

Line 178 caused

Figure 7B: strength

Figure 7C: depolymerization

line 1014, followed by ultracentrifugation and pellet (P) and supernatant (S) fractionation.

Response: We thank the reviewer to highlight the typographic errors which have now been corrected in the text and figures.

Reviewer #2 (Remarks to the Author):

The authors present exciting new insights on the role of the molecular and cellular role of dynamin 2 isoforms thorough experimentation in mouse and cell models. They show biochemical and functional differences of a muscle predominant isoform and that upregulation of the ubiquitous, not the muscle-specific isoform worsens myotubular myopathy, a fatal human muscle condition. This has important implications for the design of targeted therapies in myotubular myopathy. I have some comments for consideration:

1. The tissue expression of the muscle and the ubiquitous dynamin 2 isoforms is mainly based on RNA and protein data in mice. The authors mention briefly that similar isoforms have been found in humans according to the Gtex database. This might not be sufficient to make assumptions about the abundance of these isoforms in human muscle. I would strongly encourage the authors to specifically investigate the expression pattern in human muscle tissue.

Response: In response to the reviewer's comment, we further investigated the expression pattern of DNM2 isoforms in human skeletal muscle and the abundance of M-DNM2 isoforms in this tissue.

We performed RT-PCR and cDNA cloning from a human control muscle biopsy sample. After analysis of the clones, we confirmed the high levels of expression of M-DNM2 in human skeletal muscle with an abundance of 44% versus 56% expression of Ub-DNM2 isoforms. Thus, our data also supports the relevant expression of this isoform in human as we observed in mouse skeletal muscle (49% M-DNM2 versus 51% Ub-DNM2, Figure 1b).

We have included these novel data in **Supplementary Figure 1c** and it is now possible to compare in the same figure the specific abundance of each of the DNM2 isoforms in human and mouse skeletal muscles.

We have added the following sentence in line 137-140 of the revised manuscript: "Similar data were reported in the GTEx expression database for human (www.gtexportal.org) and we specifically confirmed this in human skeletal muscle, where M-DNM2 represented 44% of the total pool of DNM2 (Supplementary Fig. 1c)"

2. In line 279/280 "severe neonatal CNM phenotype" following the intramuscular injection of AAV into 4-week old mice is an overinterpretation and should be tuned down.

Response: We agree with the reviewer and have turned down this sentence. We meant after the intramuscular injection of Ub-DNM2, we observed in the mouse TA histopathological hallmarks found in patients with severe form of CNM, which includes fiber hypotrophy, numerous centralized nuclei, and prominent central accumulations of mitochondria.

We have rephrased the sentence, in line 289-291, as follows: "In conclusion, overexpression in WT muscle of Ub-DNM2 correlates with **histological hallmarks** of severe neonatal CNM, while expression of the M-DNM2 is linked to hallmarks found in some mild adult CNM cases."

3. In order to show the pathological consequence of the muscle and ubiquitous isoforms in muscle, the authors injected wildtype animals with the corresponding AAV constructs and showed that overexpression of the ubiquitous DNM2 isoform leads to a more profound pathology. It would be interesting to see whether similar experimentation in the XLCNM mouse (MTM1 deficient) would lead to an exacerbation of the phenotype

Response: To address if exogenous overexpression of Ub-DNM2 would aggravate *Mtm1*^{-/-} (XLCNM mouse) muscle phenotype, we performed intramuscular injections of AAV expressing Ub-DNM2 and M-DNM2 following the same protocol as in wild-type mice. TA injections were performed in 3-week-old *Mtm1*^{-/-} mice and muscles were analyzed 2 weeks post-injection as *Mtm1*^{-/-} mice starts dying at 5 weeks and we expected an exacerbation of the muscle phenotypes.

In summary, we observed a slight aggravation of the muscle phenotypes with the overexpression of both isoforms compared with *Mtm1*^{-/-} injected with empty virus, with no exacerbation of muscle hypotrophy in any case (Supplementary Fig. 11c-d). Necklaces and mis-localized nuclei were similarly increased with the expression of

both isoforms (Supplementary Fig 11e-g), while mitochondrial distribution appeared slightly more impacted with Ub-DNM2. Ub-DNM2 expression led to a slightly higher proportion of fibers with pale peripheral halo compared to M-DNM2 expression, which is a histological hallmark observed in *Mtm1*^{-/-} muscle at the latest and more severe stages of the disease (Supplementary Fig 11h-i). All these data were included in the new **Supplementary Figure 11**.

We have included the following sentence in the main text, line 284-289, referring to the main conclusion of those results: “As overexpression of different DNM2 isoforms in WT mice leads to a variable CNM pathology, we aimed to investigate if exogenous expression of DNM2 isoforms would aggravate *Mtm1*^{-/-} muscle phenotypes. AAV expressing Ub-DNM2 or M-DNM2 were injected in TA from 3-week-old *Mtm1*^{-/-} mice and resulted in a slight and similar aggravation of *Mtm1*^{-/-} histological phenotypes 2 weeks later (Supplementary Fig. 11).”

Reviewer #3 (Remarks to the Author):

This work is focused on the isoforms and functions of Dynamin 2 (DNM2) and their roles in centronuclear myopathies with a major focus on the ubiquitous (Ub-DNM2) and muscle-specific DNM2 isoforms (M-DNM2). The authors report that M-DNM2 forms large oligomers that sequester BIN1 whereas increasing Ub-DNM2 correlates with severe CNM forms and increasing M-DNM2 overexpression leads to hallmarks of mild CNM. Dnm2 splice switching from M-Dnm2 towards Ub-Dnm2 aggravates the phenotypes of severe X-linked CNM caused by loss-of-function of the MTM1. This work is focused on an uninteresting and clinically relevant topic, is well carried out and well described.

Response: We thank the reviewer for allowing us to better emphasize the interest of our study.

Following the suggestions of all reviewers, we have extensively revised this manuscript, adding *in vitro*, *in cellulo* and *in vivo* novel data. Especially, we have better dissected the differential impact of the DNM2 isoforms on the different functions of dynamin and revealed peculiar properties of these isoforms, ultimately converging to the demonstration that the muscle-specific isoform has an increased oligomerization rate, GTPase activity and GTP-induced depolymerization than Ub-DNM2 (**Figure 7**), and a decreased sensitivity to CNM mutant for membrane fission activity (**Figure 8**). The regulation of dynamin activity is a field attracting a lot of interest from researchers in biochemistry and cell biology as it is a mechanical enzyme remodeling lipid membranes. Moreover, the physiological impact of dynamins and its isoforms were barely studied, especially in the muscle tissue that is the main tissue impacted in DNM2-related genetic diseases. We also made the unexpected discovery that the ubiquitous DNM2 isoform is the one mainly responsible for the severity of phenotypes in centronuclear myopathies, while the muscle isoform is less implicated. This conclusion is sustained by our previous data and our novel data showing that a main CNM mutation is increasing the membrane fission activity of Ub-DNM2 while the M-DNM2 is resistant (**Figure 8**).

As pointed by the first reviewer we think our study “address an interesting open question in the dynamin field as the role of tissue-specific dynamin 2 splice variants and their link to the development of tissue-specific disease” and also mentioned by reviewer 2 “the authors present exciting new insights on the molecular and cellular role of dynamin 2 isoforms”. Moreover, we are bringing new insight into the CNM pathomechanism and assessing the therapeutic efficacy of targeting DNM2 by a splice switching strategy. As pointed by the second reviewer our work “has important implications for the design of targeted therapies in myotubular myopathy”. It is a clinically relevant topic as three human clinical studies have been initiated during last years including a clinical trial aiming to assess the safety, tolerability and efficacy of an antisense oligonucleotide directed against *DNM2* pan isoforms

in MTM1-CNM and DNM2-CNM patients of more than 16 years old (NCT04033159). Thus, it is clinically relevant to better understand the protein (DNM2) targeted by clinical approaches in the main tissue affected by the disease and investigate if the targeting of the muscle-specific isoform could be therapeutically efficient while decreasing potential side effects arising of the targeting of DNM2 in non-muscle tissues. Here we show, contrary to expectation, that the main target for therapy of the muscle phenotype should be the ubiquitous Ub-DNM2 isoform and not the muscle-specific M-DNM2 isoform.

My main comment is that the work is rather amorphous and that it is often unclear what the major conclusions are. At least for this reviewer the work does not well 'gel'. Perhaps a summary figure capturing the major conclusions would be helpful.

Response: The experiments suggested by reviewer 1 have helped to better link the functional experiments to the *in vivo* phenotypes as highlighted above. To summarize the main conclusions for the topics addressed (function and regulation, pathogenesis, and therapy) we have created a summary figure (**Figure 9**) as suggested by the reviewer 3. We are conscious it is a complex work, as we assessed the differential impact of the isoforms at different scales, and we expect this figure could help to link the different parts of our work and make clear what the major conclusions are. We have also revised the text to better highlight in the abstract, results and discussion the main conclusions and impact of our work in the dynamin and CNM fields.

For example, we have linked together the results of our study and major conclusions in this new paragraph in the discussion, line 453-466: "CNM mutants introduced in the Ub-DNM2 isoform have a higher propensity to self-assemble and are resistant to GTP-induced depolymerization (Gibbs *et al.* 2014, Wang *et al.* 2010, Chin *et al.* 2015, Fujise *et al.* 2021, Shan-Shan *et al.* 2020). This is especially the case of the R369W, R465W or S619L CNM mutants. Moreover, we found the addition of the CNM mutation S619L in Ub-DNM2 exacerbates its membrane fission properties in cells compared to the impact in M-DNM2, which may be already in a pre-activated conformation. Altogether, these molecular data support the Ub-DNM2 isoform is the main contributors of CNM pathogenesis as it is more sensitive to the CNM mutation, and this is in agreement with our *in vivo* data showing Ub-DNM2 overexpression in a WT and CNM (*Mtm1*^{-/-}) context has a more pathological impact than M-DNM2. Conversely, while M-DNM2 forms higher order oligomers it has a higher GTP-induced depolymerization and is less sensitive to the CNM mutant in the membrane fission assays. The fact that M-DNM2 has an opposite sensitivity to GTP-induced depolymerization than the CNM mutants, compared to Ub-DNM2, is most probably at the basis of its reduced toxicity *in vivo* (Fig. 9)."

The authors suggest that their findings support the importance of therapeutically targeting all DNM2 isoforms, but it is unclear how this could be done Can you be more specific?

Response: Our study investigated the efficacy of a splice switching strategy turning *M-Dnm2* into *Ub-Dnm2* isoforms in a XLCNM mouse model (*Mtm1*^{-/-}) and it showed this approach is not therapeutic and is leading to aggravation of CNM phenotypes. Moreover, overexpression of Ub-DNM2 is more pathogenic than M-DNM2. Altogether, these data support targeting the ubiquitous isoform for therapeutic effect rather than focusing on the muscle-specific isoform even if the disease is mainly affecting muscle. Indeed Ub-DNM2 is also expressed in muscle.

Previous strategies with therapeutic efficacy in *Mtm1*^{-/-} mice have used either AAV-shRNA complementary to *Dnm2* exon 4 or antisense oligonucleotide (ASO) against the exon 17 (Tasfaout *et al.* 2017, Tasfaout *et al.* 2018). Both approaches lead to an overall reduction of pan-*Dnm2* expression, independently of the isoform. Conversely,

our experiments showed that splice switching from M-DNM2 to Ub-DNM2, while decreasing M-DNM2, aggravates the phenotype, supporting one should not focus on M-DNM2 for therapy. Thus, only therapeutic strategies targeting Ub-*Dnm2* or pan-*Dnm2* isoforms are expected to show a therapeutic efficacy. To increase the muscle specificity of the treatment, one should not target M-DNM2 but rather engineered AAV serotypes or oligonucleotides with an increased tropism for muscle and a decreased tropism for non-muscle tissues.

Drugs inhibiting DNEM2 function or activity have never being tested in CNM mouse models, and they may not work similarly on both isoforms as our data shows functional differences between them.

Overall, our data, together with previous findings, shed lights about the therapeutic directions to treat CNM and specifically the approaches targeting DNEM2.

We have clarified this in the discussion of the manuscript including the following sentence, line 480-490: “Targeting of pan-DNM2 could be achieved by RNA interference mechanism mediated by either ASO, AAV-shRNA or siRNA complementary to *Dnm2* ubiquitous exons and leading to reduction in its overall level. Previously, we showed decreasing the overall level of pan-DNM2 in *Mtm1*^{-/-} mice rescued the disease (Cowling *et al.* 2014 and Tasfaout *et al.* 2017) (Fig. 9). Taken together, our study provides new insights about CNM therapy targeting DNEM2 and suggest, contrary to first expectation, the importance of targeting and reducing the level of pan-DNM2 isoforms and not the muscle-specific M-DNM2 isoform for an efficient therapy. Thus, increasing the muscle specificity of the treatment should not aim to target the muscle-specific DNEM2 isoform but rather to engineer AAV serotypes or oligonucleotides with an increased tropism for muscle. “

REVIEWER COMMENTS

Reviewer #1 (Remarks to the Author):

The requested revisions were carefully conducted and the new experimental data shed additional light on the mechanistic differences of Ub- and M-DNM2 as a prerequisite to understand the basis of DNEM2-mediated disease. I find this a very compelling and interesting study and recommend publication without further delay. Still, some minor text suggestions below.

Abstract

The ubiquitous Dynamin 2 (DNEM2) mechanoenzyme is a key regulator of membrane remodeling and gain-of-function mutations cause centronuclear myopathies (CNM).

Remove 'ubiquitous' here since the two isoforms are later introduced and it is not yet clear at this point that the 'ubiquitous' isoform causes disease.

In vitro, the membrane fission activity of Ub-DNEM1 was more sensitive to a CNM mutant than M-DNEM2.

Maybe better:

In a cell-based assay, a CNM-related mutation caused increased membrane scission, but only in the context of Ub-DNEM2, not M-DNEM2.

Line 66

... as higher-order helical structure

Line 95 rings

L157 –remove 'old'

L 186 – 'old' -> 'age'

Line 255 ... that Ub-DNEM2 is the main isoform linked to (remove 'the increase in')

Line 376 remove 'and' (or 'that' in the next line)

Line 378 'colocalize'

Line 442 should be 'Fig. 8'

Reviewer #2 (Remarks to the Author):

The authors have addressed all my comments including additional experimentation and data.

Response to Referees Letter

RE: Differential impact of ubiquitous and muscle dynamin 2 isoforms in muscle physiology and centronuclear myopathy (Manuscript reference: NCOMMS-21-35521B)

We thank the reviewers for the re-review of our revised manuscript based on their constructive comments and suggestions. We have carefully addressed the minor text suggestions of the reviewer 1 and provide point-by-point responses to these comments in the following sections. All modifications suggested by the reviewer has been updated in the manuscript text. We hope that this revised manuscript is suitable for publication.

REVIEWER COMMENTS

Reviewer #1 (Remarks to the Author):

The requested revisions were carefully conducted and the new experimental data shed additional light on the mechanistic differences of Ub- and M-DNM2 as a prerequisite to understand the basis of DNM2-mediated disease. I find this a very compelling and interesting study and recommend publication without further delay. Still, some minor text suggestions below.

Response: We thank the reviewer for the time to re-review our manuscript and the previously requested revisions that we carefully conducted and that shed additional light about the mechanistic insights of our present work. We also appreciate the suggested updates in the text that were corrected and implemented as indicated in following sections.

Abstract

The ubiquitous Dynamin 2 (DNM2) mechanoenzyme is a key regulator of membrane remodeling and gain-of-function mutations cause centronuclear myopathies (CNM).

Remove 'ubiquitous' here since the two isoforms are later introduced and it is not yet clear at this point that the 'ubiquitous' isoform causes disease.

Response: We updated the abstract text and removed the word ubiquitous as suggested.

Sentence in line 27-28 states as follows: "Dynamin 2 (DNM2) mechanoenzyme is a key regulator of membrane remodeling and gain-of-function mutations cause centronuclear myopathies (CNM)."

In vitro, the membrane fission activity of Ub-DNM1 was more sensitive to a CNM mutant than M-DNM2.

Maybe better:

In a cell-based assay, a CNM-related mutation caused increased membrane scission, but only in the context of Ub-DNM2, not M-DNM2.

Response: We updated abstract text following the reviewer suggestion.

Sentence in line 31-33 states as follows: "In cell-based assays, a CNM-related mutation caused increased membrane fission, only in the context of Ub-DNM2, but not M-DNM2."

Line 66

... as higher-order helical structure

Response: Line 69. Added 'order'.

Line 95 rings

Response: Line 96. Ring updated to plural 'rings'.

L157 –remove 'old'

Response: Line 158, 'old' removed and final sentence states as follows: "Dnm2ex12b^{-/-} female and male mice were phenotyped at 2wks, 8wks and 8 months (Fig. 2a)"

L 186 – 'old' -> 'age'

Response: Line 187, 'old' was substituted by 'age', thus the final sentence states as follows: "The *Mtm1*^{-ly} mouse is a faithful model for XLCNM, that displays a CNM-like histology and develops a progressive muscle weakness and atrophy from 3wks of age leading to premature death probably due to respiratory failure⁴¹".

We have also updated this through the text for consistency.

Line 255 ... that Ub-DNM2 is the main isoform linked to (remove 'the increase in')

Response: Line 255-256, sentence was updated removing 'the increase in': "Based on the above data, we hypothesized that Ub-DNM2 is the main isoform linked to the CNM pathology in the *Mtm1*^{-ly} mouse."

Line 376 remove 'and' (or 'that' in the next line)

Response: 'And' was removed in line 368, thus sentence states as follows: "BIN1 expression in COS cells induced tubule-like membrane structures originating from the plasma membrane that can recruit exogenous DNM2."

Line 378 'colocalize'

Response: Line 370, sentence was updated to: "BIN1 and DNM2 colocalize on the tubules and the vesicles^{47,48}. We observed M-DNM2 expression correlated with a slightly higher proportion of cells with vesicles or short tubules compared to Ub-DNM2 (Fig. 8a)."

Line 442 should be 'Fig. 8'

Response: We thank the reviewer for this observation as the reference to the figure number was not correct. In line 428, the mention to Fig. 7 was replaced by Fig. 8 in the Discussion section: "Finally, M-DNM2 was slightly more efficient than Ub-DNM2 to fission membrane tubules induced by BIN1 overexpression in cells (Fig. 8)."

Reviewer #2 (Remarks to the Author):

The authors have addressed all my comments including additional experimentation and data.

Response: We thank this reviewer to revise our updated manuscript and confirm all reviewer comments were addressed.